# An ABA-GA bistable switch can account for natural variation in the variability of Arabidopsis seed germination time

Katie Abley[†], Pau Formosa-Jordan[†‡], Hugo Tavares, Emily YT Chan, Mana Afsharinafar, Ottoline Leyser*, James CW Locke*

The Sainsbury Laboratory, University of Cambridge, Cambridge, United Kingdom

**Abstract** Genetically identical plants growing in the same conditions can display heterogeneous phenotypes. Here we use Arabidopsis seed germination time as a model system to examine phenotypic variability and its underlying mechanisms. We show extensive variation in seed germination time variability between Arabidopsis accessions and use a multiparent recombinant inbred population to identify two genetic loci involved in this trait. Both loci include genes implicated in modulating abscisic acid (ABA) sensitivity. Mutually antagonistic regulation between ABA, which represses germination, and gibberellic acid (GA), which promotes germination, underlies the decision to germinate and can act as a bistable switch. A simple stochastic model of the ABA-GA network shows that modulating ABA sensitivity can generate the range of germination time distributions we observe experimentally. We validate the model by testing its predictions on the effects of exogenous hormone addition. Our work provides a foundation for understanding the mechanism and functional role of phenotypic variability in germination time.

**\*For correspondence:**
ol235@cam.ac.uk (OL);
james.locke@slcu.cam.ac.uk
(JCWL)

[†]These authors contributed
equally to this work

**Present address:** [‡] Max Planck
Institute for Plant Breeding
Research, Cologne, Germany

**Competing interests:** The
authors declare that no
competing interests exist.

**Reviewing editor:** Daniel J
Kliebenstein, University of
California, Davis, United States

## Introduction

In an environment where current cues cannot be used to predict future conditions, exhibiting a range of phenotypes may promote species survival. A population can hedge its bets against an uncertain environment by maintaining genetic variation within the population (e.g. balancing selection *Delph and Kelly, 2014*) or by containing a single genotype that produces a variety of phenotypes (a diversified bet-hedging strategy *Cohen, 1966*; *Lewontin and Cohen, 1969*; *Philippi and Seger, 1989*; *Simons, 2011*). Bacteria can use diversified bet-hedging strategies to survive a range of changes of condition, including antibiotic treatments (*Balaban et al., 2004*; *Martins and Locke, 2015*) and other environmental stresses (*Patange et al., 2018*). Mechanistic studies suggest that the required phenotypic variability among genetically identical individuals is generated by genetic networks amplifying stochasticity in molecular interactions to generate a range of outputs (*Alon, 2007*; *Eldar and Elowitz, 2010*; *Viney and Reece, 2013*).

In plants, theoretical work shows that variability in the germination time of genetically identical seeds is likely to be advantageous in environments that are unpredictable (*Cohen, 1966*; *Simons, 2011*). Indeed, ecological studies have found that variability in seed germination time is correlated with environmental unpredictability (*Simons and Johnston, 2006*; *Venable, 2007*). This variability can involve germination of genetically identical seeds being spread between seasons or within a season (or a combination of both), and all these behaviours exist in wild species. Hereafter we use the term 'variability' to refer to phenotypic differences between genetically identical individuals in the same environment and 'variation' to refer to differences between genetically distinct individuals, such as natural variation between accessions.

The mechanisms underlying how seed germination is spread between seasons have been studied in detail. Seeds can enter a 'dormant' state, refractory to germination even under favourable

germination conditions (*Baskin and Baskin, 2004*; *Bewley, 1997*). Seed dormancy is a continuous variable, with genetically identical seeds having different depths of dormancy (*Finch-Savage and Leubner-Metzger, 2006*). However, the extent of dormancy for a batch of seeds is usually estimated by quantifying the percentage germination, which does not provide information about the distribution of germination times of individual seeds that germinate within a season, or experiment.

Although variability in germination times within a season has been studied in desert annuals (*Simons and Johnston, 2006*), these species are not amenable to genetic or mechanistic studies. Little is known about variability in germination time within a season in the model plant *Arabidopsis thaliana*. The extent of variability, the mechanisms that underlie it, or how related the underlying mechanisms are to those that control seed dormancy between seasons are unknown. However, there is a large body of work using percentage germination as a measure of seed dormancy. A number of quantitative genetic studies have identified loci (i.e. regions of the genome) that underlie natural variation in the extent of dormancy under different environmental conditions, with the *DELAY OF GER-MINATION* (*DOG*) loci being the first identified and some of the most characterised (*Alonso-Blanco et al., 2003*; *Bentsink et al., 2010*; *Clerkx et al., 2004*; *Footitt et al., 2020*; *Kerdaffrec and Nordborg, 2017*; *Meng et al., 2008*; *van Der Schaar et al., 1997*). The molecular mechanisms underlying germination have also been uncovered. In-depth molecular studies have shown that the decision to germinate is controlled by the balance between two hormones, gibberellic acid (GA), which promotes germination, and abscisic acid (ABA), which represses it (*Liu and Hou, 2018*). These hormones function in a mutually antagonistic manner by each inhibiting the synthesis and promoting the degradation of the other (*Liu and Hou, 2018*; *Piskurewicz et al., 2008*; *Topham et al., 2017*). Additionally, the two hormones have opposing effects on downstream transcriptional regulators that control the balance between dormancy and germination (*Liu et al., 2016*; *Piskurewicz et al., 2008*; *Shu et al., 2013*).

Pioneering modelling work has suggested that variable germination times can be generated by variation in sensitivities to germination regulators in a batch of seeds (*Bradford, 1990*; *Bradford and Trewavas, 1994*) or due to stochastic fluctuations in the regulators of ABA (*Johnston and Bassel, 2018*). Interestingly, a combination of experiment and modelling has revealed that the ABA-GA network can be described as a bistable switch due to the mutual inhibition between ABA and GA, leading to two possible states, a dormant high ABA low GA state, or a germinating low ABA high GA state (*Topham et al., 2017*). This regulatory motif can explain the observation that Arabidopsis seed germination is more effectively triggered by fluctuating temperatures than continuous cold (*Topham et al., 2017*). However, it is unclear whether different Arabidopsis genotypes generate different degrees of variability in germination times or what the role of the ABA-GA bistable switch in generating these different germination time distributions might be.

Here we set out to investigate the causes of germination time variability among genetically identical Arabidopsis seeds and explore how variability in germination time could be accounted for by the ABA-GA bistable switch. We characterise germination time distributions for natural accessions (*Vidigal et al., 2016*; *Kover et al., 2009*) (i.e. genotypically and phenotypically distinct lines, originally collected from the wild in different locations) and the MAGIC multiparent recombinant inbred line population (*Kover et al., 2009*) (previously generated by intercrossing 19 accessions, followed by multiple generations of selfing, to obtain a set of inbred lines with genomes that are mosaics of those of the parents). Using these lines, we demonstrate that there is robust natural variation in germination time variability, and that this trait shows weak coupling to mode germination time (i.e. the timing of the peak of the distribution), making it possible for these traits to vary somewhat independently. We show that variability in germination time appears to be an inherent property of each seed, rather than being dependent on differences related to seed position on the parent plant. Using quantitative trait locus (QTL) mapping, we found two loci underlying variability in germination time, with candidate genes for both loci implicated in ABA sensitivity. Testing mutants of these candidate genes provided evidence that they may regulate variability in germination time. Based on this evidence, we generate a mathematical model of the ABA-GA bistable switch that underlies germination. We show that the switch can amplify variability and account for the observed natural variation in germination time distributions. Incorporating a role for the two QTL loci in modulating ABA sensitivity into our model allows us to capture the main features of germination time distributions observed in the MAGIC lines. We validate the model by testing predictions about the effects of perturbations to ABA and GA levels in genetic lines with high or low variability. We propose that natural

variation in ABA sensitivity underlies differences in germination time distribution by affecting the degree to which the network operates in a bistable regime, as the bistable regime amplifies fluctuations to generate higher variability in germination times. Our work reveals the degree of variability in germination time in Arabidopsis as well as evidence for its genetic basis and provides a starting point for future work examining the mechanisms of phenotypic variability in seed germination time.

## Results

### Variability in seed germination time shows genetic variation in Arabidopsis

We first determined whether Arabidopsis exhibits natural variation in the variability of seed germination time. To do this, we quantified germination time distributions for 19 natural accessions and the MAGIC multiparent recombinant inbred line population, derived from those accessions (*Kover et al., 2009*). We also included 10 lines that we selected from a set of Spanish accessions as being likely to have low or high variability in germination time based on their germination time distributions over the first six days after sowing (*Vidigal et al., 2016*). We grew plants in controlled conditions for seed harvesting and collected all the seeds from three plants of each line. After a fixed period of dry storage (~30 days), we sowed a sample of each of these replicate batches of seeds in Petri dishes in controlled conditions and scored germination every day until there had been no further germination for a period of 2 weeks (see Materials and methods for further details). In these conditions, the MAGIC lines had low levels of seed dormancy (with 30 days of dry storage, 82% of lines had $\geq$ 50% germination), allowing us to quantify germination time distributions and estimate its variability. We used the coefficient of variation (CV = standard deviation/mean) of the germination time distribution as a measure of variability. We confirmed that CVs for the MAGIC parental lines remained similar over a range of lengths of dry storage period (30–60 days of dry storage), demonstrating that our results are not specific to one condition (*Figure 1—figure supplement 1A, B*).

The accessions showed a range of variabilities (*Figure 1A*). Some low CV accessions consistently germinated within 4 days, whilst higher CV lines germinated over a period of 19 days (*Figure 1A*). The MAGIC lines exhibited transgressive segregation, with greater variation in CV than the parental accessions (*Figure 1A*, compare orange and blue distributions, *Figure 1B*). The range of CVs observed in the Spanish accessions was within the range observed across all MAGIC lines (*Figure 1B*). A small number of MAGIC lines (8 out of 341 characterised) had very high CVs of germination time (>0.6) compared to the rest (*Figure 1B*), which was due to a fraction of seeds germinating very late, giving rise to bimodal distributions (e.g. M178 and M182 in *Figure 1A*).

The CV of most MAGIC lines tested was similar between repeat experiments involving independent seed harvests and sowing (*Figure 1C*) (Pearson's r = 0.88, 95% CI [0.76, 0.94] for all lines for which repeats were done). In some of the very high variability lines, the presence of very late germinating seeds was reproducible between experiments (e.g. *Figure 1C, Figure 1—figure supplement 1C, D*, M182 and M178). In other lines, very late germinating seeds were not detected in all experiments, and thus the CV was higher in some experiments than others (e.g. *Figure 1C, Figure 1—figure supplement 1E, F*, M101 and M174). Thus, although the variability in seed germination time is reproducible for most lines, for some it is possible that their CVs may have been underestimated due to a failure to detect very late germinating seeds. To check whether the level of variability in seed germination time that we obtained for a given line was related to the specific sowing conditions, we sowed selected high and low variability lines on soil and found that although the exact distributions differed slightly between Petri dishes and soil, those lines with higher variability on Petri dishes also had higher variability on soil (*Figure 1—figure supplement 1G*). Overall, our results reveal variation in germination time variability in Arabidopsis, with CVs ranging from 0.09 to 1.7 across the MAGIC lines, giving a good basis for testing the genetic mechanisms underlying this trait. The broad-sense heritability (defined as the ratio of total genetic variance to total phenotypic variance) for CV was estimated to be ~40%, which is at the upper limit of heritabilities previously measured for variability in a number of post-germination plant traits (*Hall et al., 2007*).

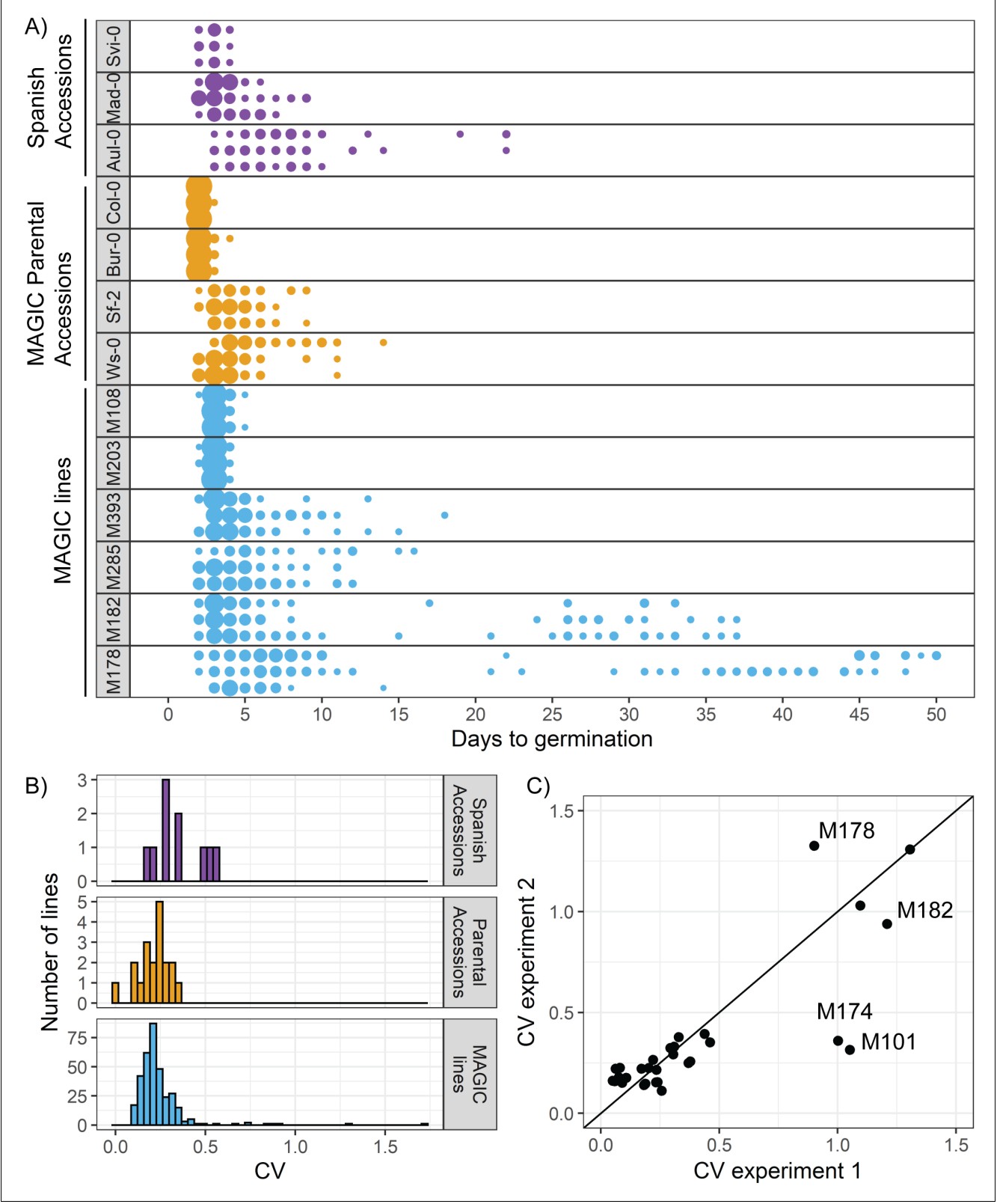

**Figure 1.** There is variation in variability in germination times in Arabidopsis. (**A**) Examples of distributions of germination time for natural accessions and MAGIC lines. Each row shows the germination time distribution of a seed batch from a different parent plant of a particular line, and colours represent whether the line is a Spanish accession (purple), MAGIC parental accession (yellow) or MAGIC line (blue). The size of the circles is proportional to the percentage of seeds sown that germinated on a given day. For the two groups of accessions (Spanish accessions and MAGIC

*Figure 1 continued on next page*

*Figure 1 continued*

parents), examples of the lowest and highest variability lines are shown. For MAGIC lines, examples are shown of low variability (top two lines); high variability, long-tailed (middle two lines) and very high variability bimodal (bottom two lines) lines. (B) Frequency distribution of coefficient of variation (CV) of germination times for 10 Spanish accessions (purple), the 19 parental natural accessions that were used to generate the MAGIC lines (orange) and 341 MAGIC lines (blue). In the majority of cases, the CV of a given MAGIC line is the mean of the CVs of three batches of seeds collected from separate parent plants. (C) CV of germination times for a subset of 32 MAGIC lines in two separate experiments. The batches of seeds for the two experiments were derived from different independently sown mother plants. The line shows y = x and is for visualisation purposes only (i.e., it does not represent a trend line). *Figure 1—figure supplement 1* shows the level of reproducibility of germination time distributions across replicates, lengths of period of dry storage and sowing conditions. *Figure 1—source data 1* contains source data for (A). *Figure 1—source data 2* contains source data for (B). *Figure 1—source data 3* contains source data for (C).

The online version of this article includes the following source data and figure supplement(s) for figure 1:

**Source data 1.** Figure 1_A_ExampleGermDistributions.
**Source data 2.** Figure 1_B_MAGICsAccessionsTraitSummaries.
**Source data 3.** Figure 1_C_MAGICExperimentComparison.
**Figure supplement 1.** Reproducibility of germination time distributions.
**Figure supplement 1—source data 1.** Figure1_figure supplement 1_A_B_MagicParentsCVvsDAR.
**Figure supplement 1—source data 2.** Figure1_figure supplement 1_CtoF_HighVarLinesReproducibility.
**Figure supplement 1—source data 3.** Figure1_figure supplement 1_G_SoilvsPlates.

## Variability in germination times is observed within single siliques

Because germination was characterised for seed samples taken from whole plants, it is possible that the high variability observed in some lines is due to different siliques (fruits) having different germination behaviours. This could arise due to differences in the ages of siliques at the time of seed harvest or due to positional effects on the parent plant. To address whether the variability in germination time that we have observed can arise independently of between-silique differences, we collected seed from samples of individual siliques from four high or very high variability lines and characterised their germination time distributions. For these lines, the full range of germination times observed in whole-plant samples was also present in seed from individual siliques (*Figure 2A, B*, *Figure 2—figure supplement 1*). This suggests that variability in seed germination time can arise independently of position or age differences between siliques.

We next hypothesised that germination time might be related to the position of the seed along the longitudinal axis of the silique. To test this, we cut siliques into halves and sowed seeds from the top halves (i.e. distal halves, furthest from the mother plant's inflorescence stem) and bottom halves separately. For the lines tested, late and early germinating seeds were produced by both halves of the siliques, with no consistent differences between the top and bottom halves of siliques in the fraction of seeds that germinated late (*Figure 2C*, *Figure 2—figure supplement 1A,B,D*). Thus, variability in germination time in the lines tested can arise independently of positional or maturation gradients within the whole plant or individual siliques. This suggests that a mechanism exists to generate differences in germination behaviour of equivalent seeds from the same silique, which is not dependent on gradients of regulatory molecules along the fruit.

## Variability is weakly coupled to modal germination time and percentage germination

To investigate which types of mechanism might underlie variability in germination time in the MAGIC population, we looked at the extent to which variability is correlated with the modal time taken to germinate and the percentage germination within the experiment. For each line, the experiment was defined as complete 2 weeks after no further seed germinated. If high CV correlates with late germination, or with low percentage germination, this would suggest that increased variability in germination times arises as a result of differences in the general regulation of germination or seed dormancy between MAGIC lines. If high CV occurs without high time to germination or low percentage germination, this would suggest that variability can be regulated somewhat independently from overall levels of seed dormancy.

We found weak correlations between CV and mode or percentage germination, with lower variability lines (low CV) tending to have a lower mode days to germination and higher percentage germination (*Figure 3A*, *Figure 3—figure supplement 1A*). Thus, some high variability lines had overall

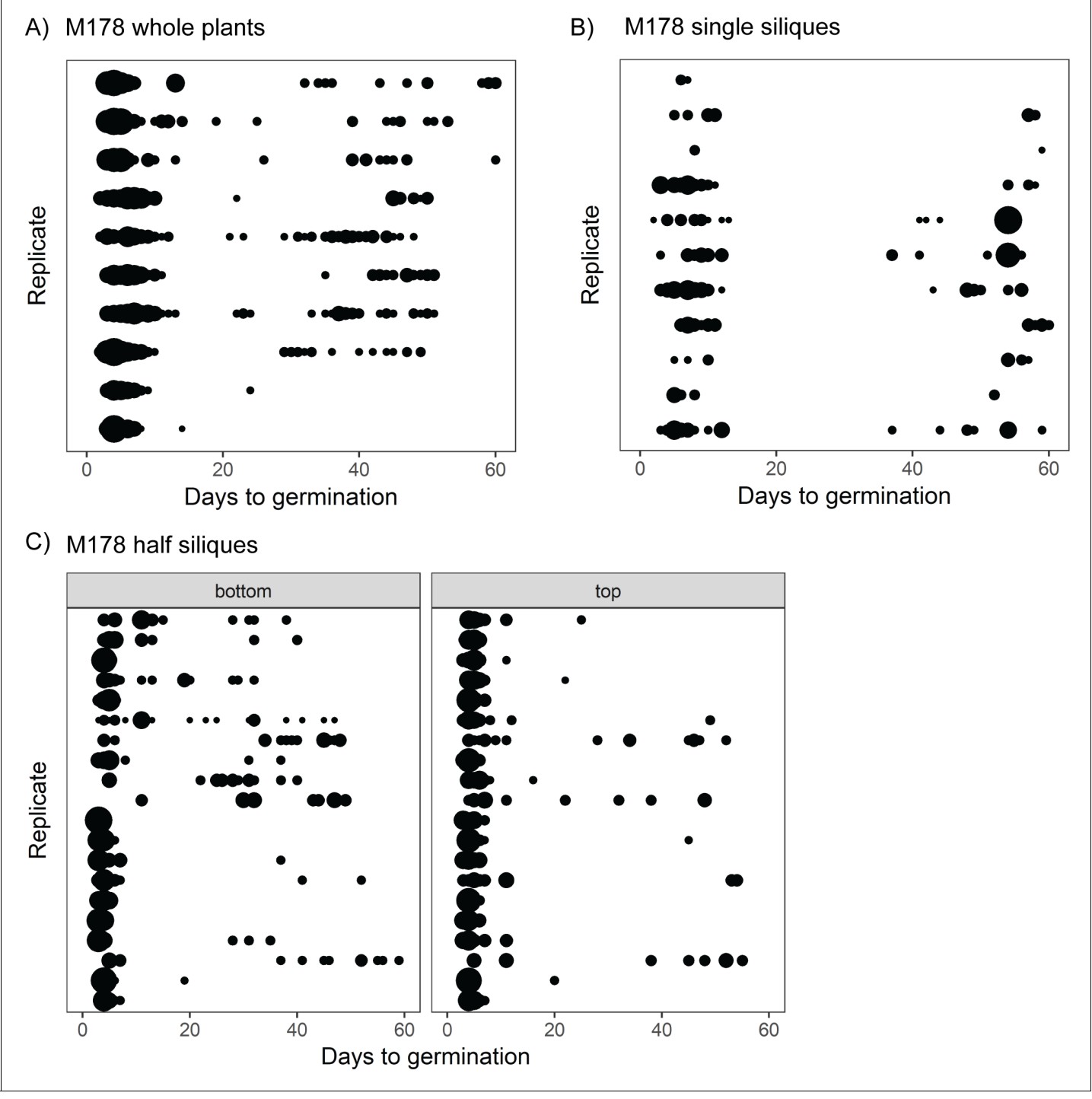

**Figure 2.** The full range of germination times can be found in individual siliques. (A) Germination time distributions for a very high variability line, M178. Each row is the distribution obtained using a sample of pooled seeds from one plant, with different rows showing data from different mother plants. (B) As for (A) but each row represents the distribution obtained using seeds from a single silique. Single siliques were randomly sampled from parent plants, and single siliques sampled from seven parent plants are represented. (C) Individual siliques were cut in half, and seeds from the top and bottom halves (distal and proximal, furthest and closest to the mother plants' inflorescence stems, respectively) were sown separately. Each row is the bottom and top half of a particular silique. Half siliques sampled from two parent plants are represented. Seeds from whole plants, single siliques and half siliques were obtained and sown in different experiments. The size of the circles is proportional to the percentage of seeds that were sown that germinated on a given day. *Figure 2—figure supplement 1* shows examples for other MAGIC lines plus an experimental repeat and statistical analysis. *Figure 2—source data 1* contains source data for (A). *Figure 2—source data 2* contains source data for (B). *Figure 2—source data 3* contains source data for (C).

*Figure 2 continued on next page*

*Figure 2 continued*

The online version of this article includes the following source data and figure supplement(s) for figure 2:

**Source data 1.** Figure2_A_M178WholePlant.
**Source data 2.** Figure2_B_M178SingleSiliques.
**Source data 3.** Figure2_C_M178HalfSiliques.
**Figure supplement 1.** Germination time distributions for whole plants and single siliques for high variability lines.
**Figure supplement 1—source data 1.** Figure2_figure supplement 1_AtoC_WholePlant.
**Figure supplement 1—source data 2.** Figure2_figure supplement 1_A_B_M182_M53_halfSiliques.
**Figure supplement 1—source data 3.** Figure2_figure supplement 1_C_M4_singleSilique.
**Figure supplement 1—source data 4.** Figure2_figure supplement 1_D_TopBottomSiliqueComparison.

later germination and lower percentage germination than low variability lines, suggesting that they were generally more dormant (*Figure 3A, D, E*, *Figure 3—figure supplement 1A*). However, there were lines that had the same mode days to germination, with very different CVs (*Figure 3A, B, C*) and vice versa, lines with the same CV showed a range of modes (*Figure 3A*). There were also lines that were very similar with respect to both percentage germination and mode days to germination, but that had very different CVs (*Figure 3—figure supplement 1*). Thus, within the MAGIC population, variability is correlated with percentage germination and modal germination time, but can be uncoupled from these traits. The same trends were observed in the natural accessions, where CV was weakly correlated with mode and percentage germination but accessions could be found with similar mode and percentage germination and different CVs (*Figure 3—figure supplement 2*).

## QTL mapping in MAGIC lines reveals two QTL underlying variability in germination time

We next performed QTL mapping on the germination data for the MAGIC lines (*Kover et al., 2009*) to investigate the genetics of germination time variability. The full set of MAGIC lines phenotyped includes lines with different types of germination time distributions. All low variability lines and most high variability lines have unimodal distributions of germination time (e.g. *Figure 1A*, M108, M203, M393 and M285). However, there are eight lines that tend to have bimodal distributions when sown on agar (e.g. *Figure 1A*, M182 and M178). As such, these lines lie at the extreme tail of the distribution of CVs, with much higher values than the other lines (*Figure 1B*). Therefore, we ran our QTL scans both with and without the bimodal lines as their extreme values may affect the QTL results disproportionately (*Figure 4A, B*).

QTL mapping for both the full set of lines and the set excluding bimodal lines revealed a region of chromosome 3 (i.e. Chr3) that accounted for ~14% of the variance in CV of germination time in the MAGIC lines used (*Figure 4A, B*). The region of significant association was broad and spanned the centromere. The tip of the peak co-located with the previously identified *DELAY OF GERMINA-TION 6* (*DOG6*) QTL, at 15.9 Mb (*Bentsink et al., 2010*; *Hanzi, 2014*). This Chr3 QTL was also associated with mean days to germination, mode days to germination and percentage germination, suggesting that this locus is a general regulator of germination time, rather than specifically affecting variability (*Figure 4—figure supplement 1*).

To investigate whether other loci explained any residual variance not explained by this major locus, we ran the QTL scans using the Chr3 QTL genotype as an additional variable (a covariate) in the model. This allows us to take into account the variance in traits that is explained by the Chr3 QTL and ask whether other regions of the genome can account for the remaining trait variance. This revealed a further putative QTL at 19.8 Mb on chromosome 5 (i.e., Chr5) associated with CV (*Figure 4B*). Unlike the Chr3 locus, this one was not significantly associated with mode or mean days to germination or percentage germination (*Figure 4—figure supplement 1*). This locus accounts for an extra 9% of the variance in CV of the MAGIC lines used. The QTL peak lies ~1.2 Mb downstream of the *DELAY OF GERMINATION 1* (*DOG1*, AT5G45830 ) gene (at 18.59 Mb) and ~1.2 Mb upstream from the *SEEDLING EMERGENCE TIME 1 (SET1)* locus (at ~21 Mb) (*Footitt et al., 2020*). The QTL scans with and without the bimodal lines were very similar for the four germination traits (CV, mean, mode days to germination and percentage germination) except the Chr5 peak was not significantly associated with CV when bimodal lines were included (*Figure 4A*, *Figure 4—figure supplement 1*).

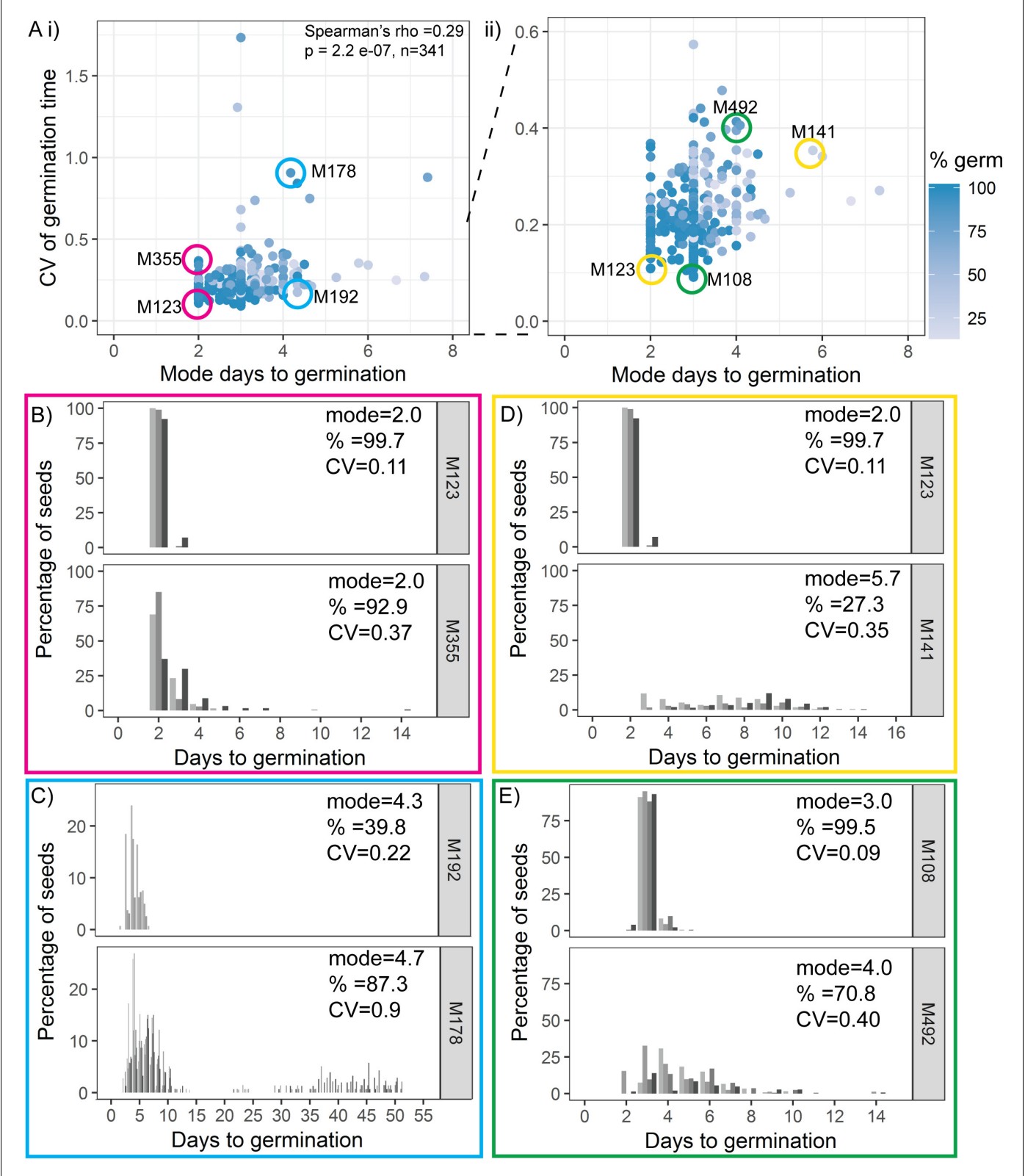

**Figure 3.** Variability is weakly coupled to modal germination time. (**A**) Scatter plots of coefficient of variation (CV) of germination time versus mode days to germination for 341 MAGIC lines. Each point is a specific MAGIC line, and in the majority of cases, the CV and mode are mean values obtained from sowing one batch of seeds from each of three separate parent plants. Each point is shaded according to the percentage germination of the line (see scale bar). Coloured circles and labels indicate lines for which examples are shown in (**B–E**). (ii) is a zoom in of (i) including only lines with CV < 0.6.

*Figure 3 continued on next page*

*Figure 3 continued*

Spearman's correlation for the full set of 341 MAGIC lines is indicated in (i). (B–E) Distributions of germination times for pairs of MAGIC lines. The colour of the box matches the coloured circles in (A). Lower CV lines are shown on top. Grey-coloured bars show the germination time distribution of seed batches from replicate mother plants. (B, C) Exemplar lines with the same mode days to germination but different CVs of germination time. (D, E) Lines that have different CVs and different mode days to germination. For each line, the mode days to germination, final percentage germination and CV of germination time are shown. Note that the x-axis scale differs between plots. *Figure 3—figure supplement 1* shows the relationship between CV and percentage germination for MAGIC lines. *Figure 3—figure supplement 2* shows relationships between CV, mode and percentage germination for natural accessions. *Figure 3—source data 1* contains source data for (A). *Figure 3—source data 2* contains source data for (B–E).

The online version of this article includes the following source data and figure supplement(s) for figure 3:

**Source data 1.** Figure3_AllMAGICsTraitSummaries.

**Source data 2.** Figure3_MAGICIndividualLinesGermPerDay.

**Figure supplement 1.** Variability is weakly coupled to percentage germination.

**Figure supplement 2.** The relationship between coefficient of variation (CV), mode days to germination and percentage germination in natural accessions.

**Figure supplement 2—source data 1.** Figure3_figure supplement 2_A_B_AccessionsTraitSummaries.

**Figure supplement 2—source data 2.** Figure3_figure supplement 2_C_AccessionsCt_Mad_GermPerDay.

We next estimated the effects of particular accession haplotypes at the two QTL on the different germination traits (*Figure 4—figure supplement 2*). For the Chr3 QTL, there was a relatively strong negative correlation between haplotypic effects on CV and percentage germination, and a positive correlation between effects on CV and mode (*Figure 4—figure supplement 2B, C*). This supports the conclusion that this QTL is a general regulator of seed germination time. For the Chr5 QTL, there was a negative correlation between haplotypic effects on CV and percentage germination, but there was no correlation between their effects on CV and on mode (*Figure 4—figure supplement 2B, C*).

To confirm the effect of the Chr5 QTL on CV in an independent experiment, we used an F2 bulked-segregant mapping approach in a cross between two accessions (Col-0 and No-0) predicted to have haplotypes in this genomic region with different effects on CV (*Figure 4—figure supplement 2A*, Chr5 panel). We performed whole-genome sequencing on pools of F2 plants that germinated late, at the right tail of the F2's germination time distribution, and were therefore predicted to be enriched for the No-0 haplotype at ~20 Mb on Chr5, promoting high variability. We compared their sequences to those of a pool of early germinating F2 plants (*Figure 4C*; for details of pools, see *Figure 4—figure supplement 3*). The results independently verified that a locus at ~20 Mb of Chr5 has an influence on variability. In this experiment, the peak of association was located at 18.6 Mb on Chr5, which overlaps precisely with the *DOG1* gene (*Figure 4C*). We also quantified germination traits of the F3 offspring of F2 plants that themselves germinated early or late. This showed that late germinating F2 plants produced seeds with higher CVs of germination time, lower percentages of germination and similar average germination times compared to seeds of early germinating F2 plants (*Figure 4—figure supplement 4*).

In summary, we have shown that at least two loci contribute to variability in seed germination time in the MAGIC lines (Chr3, ~16 Mb and Chr5, ~18.6/19.8 Mb). Consistent with a correlation between CV, mode days to germination and percentage germination in the MAGIC lines, the main QTL on Chr3 has correlated effects on all these three traits. The locus at ~19 Mb on Chr5 appears to affect variability most strongly.

The Chr5 peak obtained in the bulk segregant mapping overlaps with the *DOG1* gene known to play a role in seed dormancy level. The peak obtained in the QTL mapping is slightly shifted and lies equidistant between *DOG1* and the nearby *SET1* locus (at ~21 Mb) which affects dormancy levels in the field in response to environmental conditions. Consistent with a role for this region of Chr5 in seed dormancy in the MAGIC lines, its haplotypic effects on CV and on percentage germination were negatively correlated (*Figure 4—figure supplement 2B*, Chr5 panel). Additionally, our Col-0 × No-0 F2 and F3 analysis suggested that seeds from plants enriched for the No-0 haplotype at this locus (which is associated with high CV) had a lower percentage germination than seeds from plants enriched for the low CV Col-0 haplotype (*Figure 4—figure supplement 4*). However, perhaps surprisingly, this locus was not significantly associated with percentage germination in the QTL mapping (*Figure 4A, B*). This may be because, unlike the Cvi accession that was used originally to map both *DOG1* and *SET1* loci (*Alonso-Blanco et al., 2003*; *Footitt et al., 2020*), the accessions used to

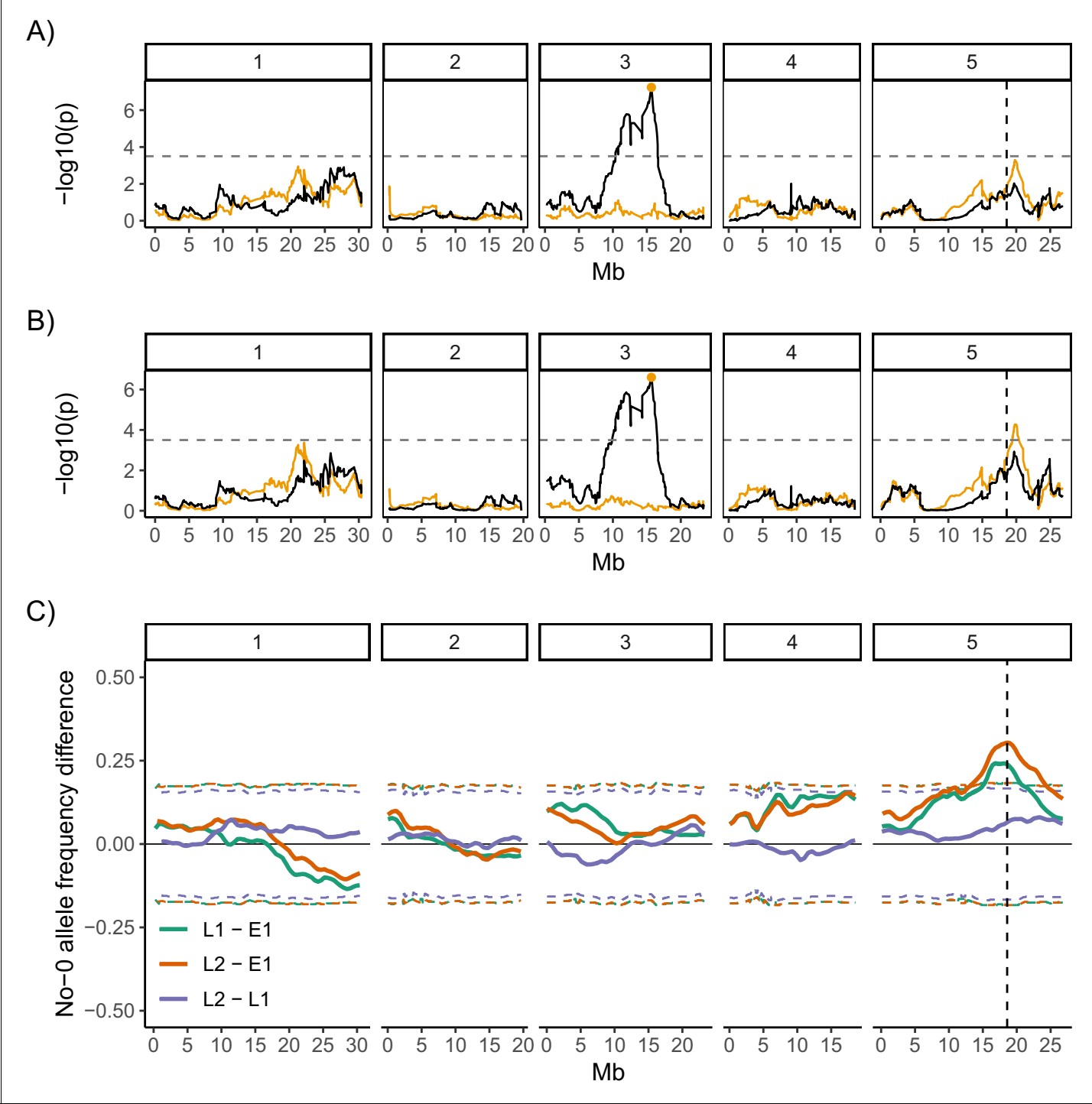

**Figure 4.** Quantitative trait locus (QTL) and bulk segregant mapping reveals two QTL underlying coefficient of variation (CV) of germination time. (A, B) Manhattan plots showing the QTL association results for each single nucleotide polymorphism (SNP) marker individually (black line) and for each marker when the Chr3 QTL SNP marker was added as a covariate (i.e. an additional variable) in the model (orange line). The orange line shows the variation in CV that is accounted for by each SNP across the genome when the variation that is explained by the Chr3 QTL SNP marker (orange point) is accounted for by adding it to the model as a covariate. The y-axis shows the p-values for the 1254 markers used, on a negative $\log_{10}$ scale, such that higher peaks indicate a stronger association between the region of the genome and CV. The numbered panels represent the five chromosomes of Arabidopsis. The horizontal dashed line shows a 5% genome-wide threshold corrected for multiple testing (based on simulations in *Kover et al., 2009*). The vertical dashed line indicates the *DOG1* gene. (A) is for the full set of 341 MAGIC lines that was phenotyped and (B) excludes the eight bimodal lines with very high CV. *Figure 4—figure supplement 1* shows QTL mapping for mean and mode days to germination and percentage germination. *Figure 4—*

*Figure 4 continued on next page*

*Figure 4 continued*

*figure supplement 2* shows estimated effects of accession haplotypes on CV, mode and percentage germination. (**C**) Mapping QTL by bulk-segregant analysis using whole-genome pooled sequencing of F2 pools from a Col-0 × No-0 cross. One early and two late germinating F2 pools were sequenced. The plot shows the No-0 allele frequency differences between pairs of pools indicated in the legend (*Figure 4—figure supplement 3* shows details of pool selections; E1, "early pool"; L1, "late one pool"; L2, "late two pool"). The horizontal dashed lines indicate the 95% thresholds based on simulating the null hypothesis of random allele segregation, taking into account the size of the sampled pools and the sequencing depth at each site (*Magwene et al., 2011*; *Takagi et al., 2013*). Positive values above the top line indicate enrichment for No-0 alleles, while negative values below the bottom line indicate enrichment for Col-0 alleles. As predicted, late germinating pools were enriched for the No-0 haplotype in the region of the Chr5 QTL. Here, the peak of association overlaps with the *DOG1* gene (dashed vertical line). *Figure 4—figure supplement 4* shows germination phenotypes of F3 seeds from Col-0 x No-0 F2 plants that themselves germinated early or late.

The online version of this article includes the following source data and figure supplement(s) for figure 4:

**Figure supplement 1.** Quantitative trait locus mapping for germination traits, with and without bimodal MAGIC lines.
**Figure supplement 2.** Accession-specific quantitative trait locus (QTL) effects on coefficient of variation (CV), mode and percentage germination.
**Figure supplement 3.** Germination time distributions and DNA-seq pools of Col-0 × No-0 F2.
**Figure supplement 3—source data 1.** Figure4_FigureSupplement3_A_B_ColNoMappingF2_GermDistributions.
**Figure supplement 4.** Germination phenotypes of F3 seeds from Col-0 × No-0 F2 parent plants that themselves germinated early or late.
**Figure supplement 4—source data 1.** Figure4_FigureSupplement4_ColNoF3_GermDistributions.
**Figure supplement 5.** Germination time distributions and abscisic acid (ABA) dose responses in quantitative trait locus candidate gene mutants.
**Figure supplement 5—source data 1.** Figure4_FigureSupplement5_QTLcandidateMutantsGerm.

---

generate the MAGIC lines have relatively weak dormancy and may not carry alleles in this region that promote dormancy sufficiently strongly to be detected for percentage germination in the QTL mapping.

## Effects of QTL candidate genes on seed germination time variability

There is evidence to suggest that the best candidate genes underlying our identified loci influence ABA sensitivity. The effect of the *DOG6* locus overlapping our Chr3 QTL is proposed to be caused by the *ANAC060* gene (*Hanzi, 2014*), which influences ABA sensitivity in seedlings, directly binds to the promoter of the ABA-responsive transcription factor, *ABA INSENSITIVE 5* (*ABI5*), and can down-regulate expression of both *ABI4* and *ABI5* (*Li et al., 2014*; *Yu et al., 2020*). Coincidentally, the two candidate genes for the Chr5 locus are closely related in function. The *SET1* locus in this region is hypothesised to be caused by the *ABA-HYPERSENSITIVE GERMINATION 1* (*AHG1*) gene (*Footitt et al., 2020*). *AHG1* is a type 2C protein phosphatase (PP2C) that inhibits ABA signalling via dephosphorylating class II SNF1-related protein kinase 2 (SnRK2), which promote seed dormancy by activating ABA-responsive transcriptional changes (*Liu and Hou, 2018*; *Née et al., 2017a*; *Nishimura et al., 2007*, *Nishimura et al., 2018*). DOG1 has been shown to directly bind to AHG1, independently from ABA, and inhibit its function, thus allowing DOG1 to inhibit germination via the ABA pathway (*Carrillo-Barral et al., 2020*; *Née et al., 2017a*; *Nishimura et al., 2018*). Mutants of all three candidate genes have altered ABA sensitivity: *anac060* mutant seedlings and *ahg1* mutant seeds have increased ABA sensitivity (*Yu et al., 2020*; *Li et al., 2014*; *Nishimura et al., 2007*), while *dog1* mutant seeds have decreased ABA sensitivity (*Née et al., 2017a*).

We tested these mutants for their effects on germination time distributions and found that the *dog1-3* mutant in the Col-0 background (*Bentsink et al., 2006*) consistently had reduced CV of germination time, and reduced mean and mode days to germination compared to the wild type (*Figure 4—figure supplement 5A*). Since in Col-0 all seeds normally germinate within 3 days in seed batches stored for 30 days, we also did this experiment in seed batches stored for a shorter period of time (5 days after harvesting [DAH]). In this case, in Col-0 there were later germinating seeds that were not present in the *dog1-3* mutant, making the effect of the *dog1-3* mutant allele more apparent. We also obtained a *dog1* T-DNA insertion mutant in the No-0 accession background (*Kuromori et al., 2004*) since the No-0 haplotype of the Chr5 QTL locus is predicted to be associated with high variability in the MAGIC lines (*Figure 4—figure supplement 2A*). The *dog1* mutant in the No-0 background showed a similar phenotype to *dog1-3* in Col-0, with reduced CV, mean and mode (*Figure 4—figure supplement 5C*). Consistent with previously published work (*Née et al., 2017a*), we observed reduced ABA sensitivity of the *dog1* mutants in the Col-0 and No-0 backgrounds in our germination assays, with higher levels of exogenous ABA required to observe

a change in the germination fraction (*Figure 4—figure supplement 5B, D*). On the other hand, the *ahg1-5* mutant in the Col-0 background (*Née et al., 2017a*) showed a slight increase in CV of germination time, which was associated with an increase in ABA sensitivity (*Figure 4—figure supplement 5E, F*). Thus, both of these candidate genes for the Chr5 locus have an effect on the CV of germination time which is consistent with their altered ABA sensitivities.

For the *anac060* mutant in the Col-0 background (*Li et al., 2014*), we observed a slight increase in CV, with an increase in mode and decrease in percentage germination compared to Col-0 in seed sown 3 DAH (*Figure 4—figure supplement 5G*, 3 DAH), but did not see a convincing phenotype in seeds that were stored for 30 days prior to sowing (i.e. in the same conditions as the MAGIC lines used for QTL mapping) (*Figure 4—figure supplement 5G*, 30 DAH). We observed a weak tendency towards an increase in ABA sensitivity in the *anac060* mutant, but this was not as striking as that reported in seedlings (*Li et al., 2014*). Thus, it is possible that a gene other than *ANAC060* underlies the Chr3 QTL, or if it is responsible, then its effect on germination may depend on the genetic background or on the specific alleles present in the populations we studied.

Overall, our results support the hypothesis that the candidate genes underlying the Chr5 QTL peak could influence variability through an effect on ABA sensitivity and suggest that this could also be the case for the *ANAC060* candidate gene for the Chr3 QTL peak.

## A stochastic model of the ABA-GA bistable switch can account for the observed genetic variation in germination time distributions

The hypothesis that natural variation in germination time distributions is caused by differences in ABA sensitivity raises the question of how differences in ABA sensitivity between lines could affect their levels of variability in germination time. To answer this, we built a simplified mathematical model of the core ABA-GA network that governs germination time (*Liu and Hou, 2018*; *Figure 5A*). We reasoned that it was necessary to include both ABA and GA in the model since the decision to germinate is governed by the relative levels of the two hormones (*Née et al., 2017b*; *Shu et al., 2016b*), and both converge to regulate the expression of a common set of transcription factors that control seed dormancy and germination (*Liu and Hou, 2018*; *Piskurewicz et al., 2008*; *Shu et al., 2016a*). A previous modelling study that solely considered ABA regulation has proposed that stochastic fluctuations in the regulation of ABA can generate variability in germination times (*Johnston and Bassel, 2018*) and the ABA-GA network has been modelled previously to account for germination decisions (*Topham et al., 2017*). However, the ability of the ABA-GA network to generate variability in germination time has not been explored.

Our mathematical model captures the relationships between the hormones ABA and GA and the key transcriptional regulators that act as inhibitors of germination, such as DELLAs, ABI4 and ABI5 (*Ariizumi et al., 2008*; *Liu et al., 2016*; *Piskurewicz et al., 2008*; *Shu et al., 2016a*; *Tyler et al., 2004*). We represent these germination inhibitors as one factor, called Integrator. We model the net effects of ABA and GA on the germination inhibitors by assuming that the production of Integrator is promoted by ABA, and that its degradation is promoted by GA (*Figure 5A*; *Ariizumi et al., 2008*; *Liu et al., 2016*; *Piskurewicz et al., 2008*; *Shu et al., 2016a*; *Tyler et al., 2004*). The germination inhibitors are known to feed-back to influence GA and ABA levels through effects on their biosynthesis or catabolism (*Ko et al., 2006*; *Oh et al., 2007*; *Piskurewicz et al., 2008*; *Shu et al., 2016a*; *Shu et al., 2013*). This feedback is represented in the model by assuming that Integrator promotes the production of ABA (*Ko et al., 2006*; *Zentella et al., 2007*) and inhibits the production of GA (*Shu et al., 2013*; *Oh et al., 2007*). To capture the inhibitory effect of the DELLAs, ABI4 and ABI5 on germination, we assume that in each seed the Integrator level must drop below a threshold for germination to occur. Finally, we include a factor, Z, to simulate a light-induced increase in GA production rate upon sowing (*Derkx and Karssen, 1993*; *Oh et al., 2007*, *Oh et al., 2006*). Full details and justifications of the model assumptions are provided in the Materials and methods section.

The model behaves as a mutual inhibition circuit (GA inhibits Integrator and vice versa) and a mutual activation circuit (ABA promotes integrator and vice versa) coupled by the Integrator (*Figure 5A*). Overall, this constitutes a double positive feedback loop that can act as a bistable switch, where there are two stable steady state solutions: low Integrator, low ABA and high GA resulting in germination; or high Integrator, high ABA and low GA resulting in no germination (*Figure 5—figure supplement 1H*). We hypothesised that variability in germination time is generated from stochastic fluctuations in the dynamics of the underlying gene regulatory network. To model

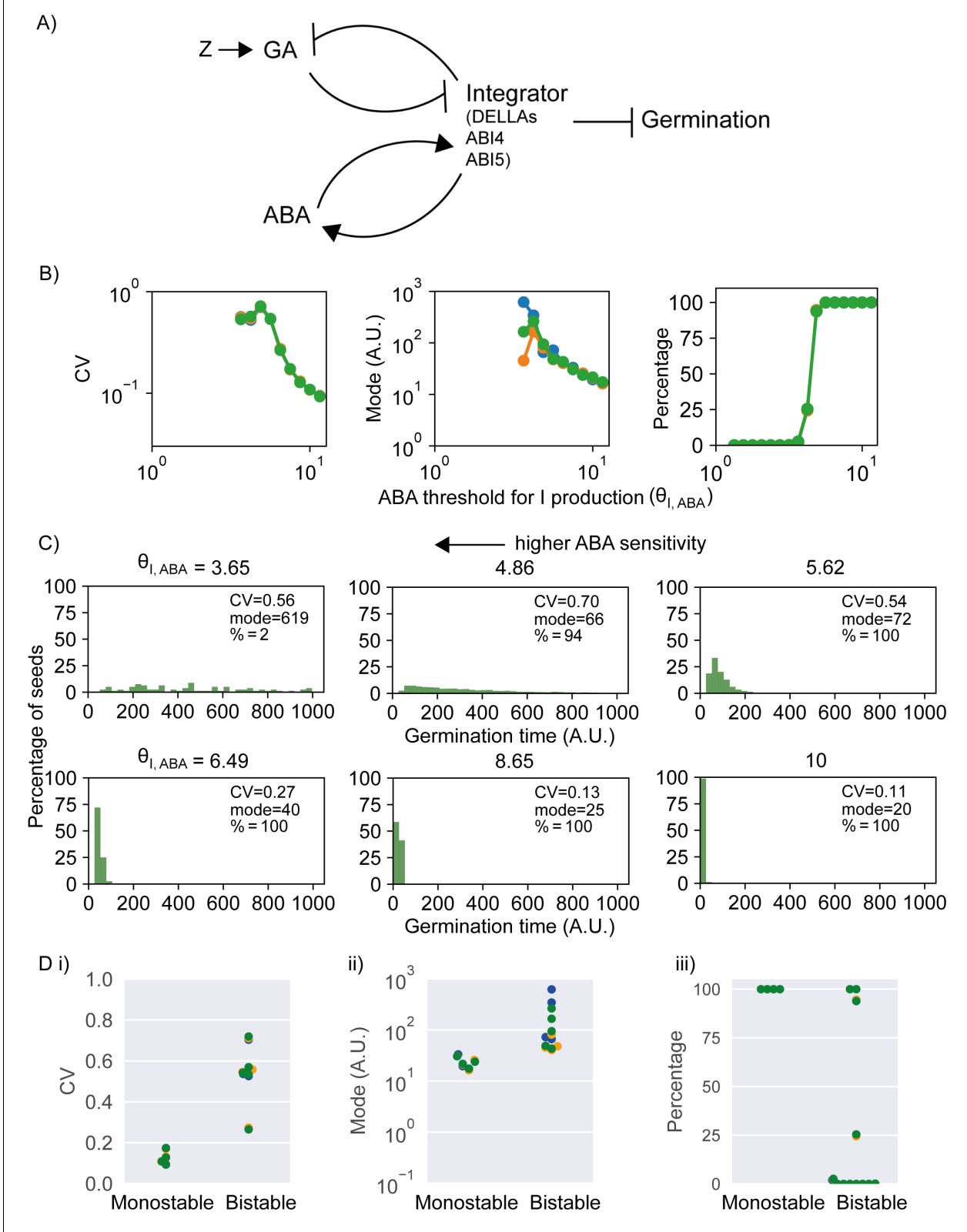

**Figure 5.** Model of the abscisic acid–gibberellic acid (ABA-GA) bistable switch and effect of ABA sensitivity parameter on germination traits. (**A**) Model scheme of the ABA-GA network. Normal arrows represent effective promotion and blunt arrows represent effective inhibition. We represent the inhibitors of germination – DELLAs, ABI4 and ABI5 – as one factor, called Integrator, which we assume must drop below a threshold for germination to occur. We assume that ABA promotes the production of Integrator and that GA promotes its degradation. Integrator is assumed to promote ABA

*Figure 5 continued on next page*

*Figure 5 continued*

production and inhibit GA production. A factor, Z, increases upon sowing and promotes GA production. *Figure 5—figure supplement 1* provides information on the dynamics of the model. (B) Effects on coefficient of variation (CV), mode and percentage germination of simulated germination time distributions as the ABA threshold for Integrator (I) production parameter values are changed. This parameter is inversely correlated with sensitivity of Integrator to ABA. Each panel shows the results of three different runs of stochastic simulations on 4000 seeds. (C) Simulated germination time distributions for six values of the ABA threshold for Integrator production parameter, showing positively correlated changes in CV and mode. The arrow indicates increasing sensitivity of Integrator production to ABA towards the top left. (D) CV, mode and percentage germination in bistable and monostable regions of the model parameter space after the rise in GA production (see *Figure 5—figure supplement 1* for details of monostable and bistable regimes). See Materials and methods for regions of the parameter space that we exclude from these plots because they are considered less biologically relevant. Colours in (B) and (D) represent different runs of stochastic simulations. See Materials and methods for further details on parameters and numerical simulations.

The online version of this article includes the following figure supplement(s) for figure 5:

**Figure supplement 1.** Dynamics of the components of the abscisic acid–gibberellic acid (ABA-GA) model in monostable and bistable regimes.

**Figure supplement 2.** Effect of model parameters on germination traits.

**Figure supplement 3.** Simulated germination time distributions illustrating the effects of parameter value changes.

**Figure supplement 4.** Exploring the effects of model parameters on coefficient of variation (CV), mode and percentage germination.

**Figure supplement 5.** Coefficient of variation (CV), mode of germination times and percentage germination in bistable and monostable regions of the model after the rise in GA production.

these stochastic fluctuations, we adopt the chemical Langevin equation formalism (see Materials and methods), which takes into account the intrinsic stochasticity of the chemical reactions happening within the cell throughout time (*Adalsteinsson et al., 2004*; *Gillespie, 2000*).

Although the model is a simplified representation of the interactions between GA and ABA, it can make predictions concerning network behaviour. To investigate the effect of ABA sensitivity on germination time distributions in the model, we varied the parameter governing the ABA threshold for Integrator production, which is inversely correlated with the sensitivity to ABA (see Materials and methods). Varying this parameter can cause correlated effects on CV and mode (*Figure 5B*) and can account for a range of germination time distributions, from less variable (i.e. more peaked), to more variable (i.e. long tailed) (*Figure 5C*), that qualitatively match the range of germination time distributions we observe experimentally (*Figure 1A*, *Figure 3*). As the sensitivity of integrator to ABA increases (lower values of ABA threshold for integrator production), the germination time distributions are more long tailed with a higher CV and mode (*Figure 5C*). For very high sensitivities, the germination time distributions become flat.

Changes to the sensitivity of Integrator to ABA affect CV by influencing the stability of the non-germination steady state and whether it exists following sowing. In our modelling approach, we assume that all seeds of all lines exist in a stable non-germinating state prior to sowing, and, as mentioned above, we assume that sowing (i.e. exposure to water and light) causes a rise in the rate of GA production, which enables this situation to change. After this rise in GA production, the model may operate in a monostable regime where germination is the only possible state (i.e. the non-germination steady state is lost) (*Figure 5—figure supplement 1A, B*), or a bistable regime, where seeds may either germinate or not (*Figure 5—figure supplement 1H, I*). With low sensitivity to ABA, the model is monostable after sowing, with the high GA, low ABA, low Integrator (germination) steady state the only stable state after the sowing-induced rise in GA production (*Figure 5—figure supplement 1A, B*). This means that all seeds switch rapidly from their initial non-germinating state (as it disappears upon the rise in GA production) into the germination state (*Figure 5—figure supplement 1C–G*). With higher ABA sensitivity, the model is bistable after sowing and the rise in GA production, with a low GA, high ABA, high integrator (non-germination) steady state in addition to the germination steady state both existing after sowing (*Figure 5—figure supplement 1H, I*). In this bistable scenario, after the rise in GA production, seeds can remain in the non-germination steady state for some time but are driven to transition to the germination steady state by stochastic fluctuations, which results in variable germination times and a later average germination time (*Figure 5—figure supplement 1J–N*). Hence, the bistable regime is associated with an increase in CV and mode of germination time (*Figure 5D*). Thus, the coupled variation in CV and mode of MAGIC line germination time distributions can be at least partly accounted for by variation in the parameter

controlling the sensitivity of integrator production to ABA: as the sensitivity to ABA increases, the model becomes bistable, with later and more variable germination times.

Additionally, once within the bistable regime, increases in ABA sensitivity increase the stability of the non-germination steady state (*Figure 5—figure supplement 1O–U* compared with *Figure 5—figure supplement 1H–N*). This can result in an increasing proportion of seeds remaining in this state and not germinating (decreasing percentage germination) and can also further increase the variability of germination time of the seeds that do germinate (*Figure 5—figure supplement 1Q–U* compared to *Figure 5—figure supplement 1J–N*). Hence, we hypothesise that natural variation between MAGIC lines in the variability of germination time could be due to differences in ABA sensitivity that cause (i) the ABA-GA network to operate in different regimes (i.e. monostable versus bistable) in different lines and (ii) differences between lines in the stability of the non-germination steady state.

It should be noted that the model can also show other types of dynamical behaviour in addition to the regimes described. These include when the rise in GA production is not enough to enable the existence of a stable germination state (i.e. in this case, the integrator stable state is above the germination threshold), but germination still occurs, being driven by stochastic fluctuations. However, we consider the monostable and bistable regimes that we describe the most biologically relevant (see Materials and methods for a full description of possible behaviours and a discussion of their biological relevance).

We next performed parameter screens for other parameters in the model and investigated their effect on CV and mode. We were interested in whether parameters other than that governing ABA sensitivity could account for the coupling between CV and mode, and whether other parameters could have decoupled effects on CV and mode, to explain the weak coupling between these traits in the MAGIC lines. Specifically, we varied the basal production and degradation rates of ABA, GA and Integrator, the parameters governing the sensitivity of the interactions between the three factors, as well as the level of noise in the system. We performed 2D parameter explorations to check the effect of varying a given parameter for a range of values of a second parameter, to ensure that the behaviours observed were robust across a range of parameter sets (*Figure 5—figure supplement 4*).

We found that the model could capture multiple possible relationships between the CV and mode of germination time distributions, with the nature of the relationship changing depending on which parameter was being varied. A number of parameters showed positively correlated effects on CV and mode (e.g. *Figure 5—figure supplement 2A*), although the strength of this correlation varied depending on the region of parameter space and the parameter being changed (*Figure 5—figure supplement 4*). We confirmed that these relationships between CV and mode were not due to changes in the percentage of seeds germinating as they were observed in cases where percentage germination remained constant (*Figure 5—figure supplement 2A*). A positive correlation between effects on CV and mode was observed for parameters controlling the rates of basal production and degradation of GA, ABA and Integrator (*Figure 5—figure supplement 4A–C*) as well as for those controlling the GA-dependent degradation rate of the Integrator (*Figure 5—figure supplement 4D*). Positive correlations between the effects on CV and mode were also observed for the sensitivity of ABA production to Integrator levels (we define sensitivity as the inverse of the Integrator threshold for promotion of ABA production and use the equivalent definition for all subsequent sensitivity parameters; see *Figure 5—figure supplement 4E*). Thus, multiple other parameters in addition to that governing ABA sensitivity (*Figure 5B, C*, *Figure 5—figure supplement 4F, G*) could account for the positive correlation between CV and mode of germination time distributions in the MAGIC lines. In most cases, low percentage germination tends to be associated with high CV and mode while high percentage germination tends to be associated with low CV and mode, similar to the correlations observed within the MAGIC lines (e.g. see *Figure 5B, C*, *Figure 5—figure supplement 4A–C*, *Figure 3—figure supplement 1A*). There are also regions of parameter space where high percentage germination is associated with high CV and high mode (*Figure 5—figure supplement 4C*), which is consistent with observation that the germination traits can be uncoupled in the MAGIC lines.

Dependent on the area of the parameter space, specific parameters could show different effects on CV and mode. For example, although varying the ABA sensitivity parameter tends to have positively correlated effects on CV and mode (*Figure 5*), for some regions of parameter space it does not affect these traits (*Figure 5—figure supplement 4F, G*). Additionally, some parameters

showed both correlated and anti-correlated effects on CV and mode and some generated decoupled changes in CV and mode. For example, for the parameter controlling basal GA production and that controlling the rate of Integrator production, in addition to the positively correlated effects on CV and mode described above, anti-correlated effects on these traits were also observed in other regions in parameter space (*Figure 5—figure supplement 4B, C*). For some areas of parameter space, anti-correlated effects on CV and mode occurred when modulating the sensitivity of the Integrator to GA-promoted degradation (*Figure 5—figure supplement 2B*, *Figure 5—figure supplement 3B*, *Figure 5—figure supplement 4F, H*). In other regions of the parameter space, varying this parameter caused larger changes in mode than in CV (*Figure 5—figure supplement 2C*, *Figure 5—figure supplement 3C*). Somewhat decoupled changes in CV and mode were also observed when varying the parameter that controls the level of noise in the system, such that, for some regions of parameter space, reductions in noise decreased the CV while maintaining a relatively constant mode and percentage germination (*Figure 5—figure supplement 2D*, *Figure 5—figure supplement 3D*, *Figure 5—figure supplement 4G, H*). Thus, the model can capture complex relationships between different germination traits.

To understand further how differences in CV and mode of germination time distributions are generated by the model, we looked across all the parameter screen results to see how CV and mode varied as the model switched between the situation where the model is monostable after the rise in GA production (e.g. *Figure 5—figure supplement 1A–G*) and the situation where it is bistable following this rise in GA production (e.g. *Figure 5—figure supplement 1H–U*). We found that across a range of parameter sets, both modes and CVs of germination times tend to be higher when bistability rather than monostability occurs after the sowing-induced rise in GA production (*Figure 5—figure supplement 5*). Thus, in general, the model predicts that high variability in germination time is associated with the ABA-GA network operating in the bistable regime.

## Exogenous ABA and GA addition validates the model predictions

To generate testable predictions, we next sought to understand how the model behaves when the levels of ABA and GA are varied through exogenous addition. To simulate ABA and GA addition to high and low variability MAGIC lines, we represent these two different classes of lines with a difference in the parameter controlling the sensitivity of Integrator production to ABA. The low variability lines are represented by low ABA sensitivity, with the model operating in the monostable regime with a more peaked germination time distribution. The high variability lines are represented by high ABA sensitivity, with the model operating in the bistable regime and having a longer tailed germination time distribution.

Although here we chose to model low variability lines as being in the monostable regime, some parameter sets in the bistable regime can also be associated with relatively low CV (*Figure 5—figure supplement 5*), hence it is possible that some low variability Arabidopsis lines operate within the monostable regime whilst others operate within the bistable regime. This would be consistent with the experimental observation that the MAGIC lines with lowest variability have 100% germination (*Figure 3—figure supplement 1Aii*) suggesting that they operate within the monostable regime, while there are also low variability MAGIC lines that have lower germination percentages, suggesting that they operate within the bistable regime (*Figure 3—figure supplement 1A*).

The model predicts that starting from germination time distributions with low or high variability, increasing concentrations of exogenous ABA will initially increase the CVs of the germination time distributions (*Figure 6Ai*), causing long-tailed distributions to emerge (*Figure 6—figure supplement 2A*). This is because the addition of exogenous ABA stabilises the high Integrator, high ABA, low GA non-germination state, requiring stronger fluctuations to allow germination (*Figure 6—figure supplement 3A*). At higher concentrations of exogenous ABA, the germination time distribution becomes flattened into a seemingly uniform distribution with a high mean and therefore lower CV (*Figure 6Ai, ii*, *Figure 6—figure supplement 2A*, [*ABA*]exo = 2.5). In this situation, the mode shows an increase with some noticeable fluctuations due to the flattened germination time distribution, where mode is less well defined (*Figure 6Aiii*). At high enough levels of exogenous *ABA*, the time to achieve the low integrator state can become larger than our chosen final simulation time; seeds exhibiting this behaviour are not considered as germinated, emulating the finite time window during which the germination scoring is performed in this experiment (see Materials and methods). Hence, the increase of exogenous ABA also reduces the percentage of germinated seeds (*Figure 6Aiv*).

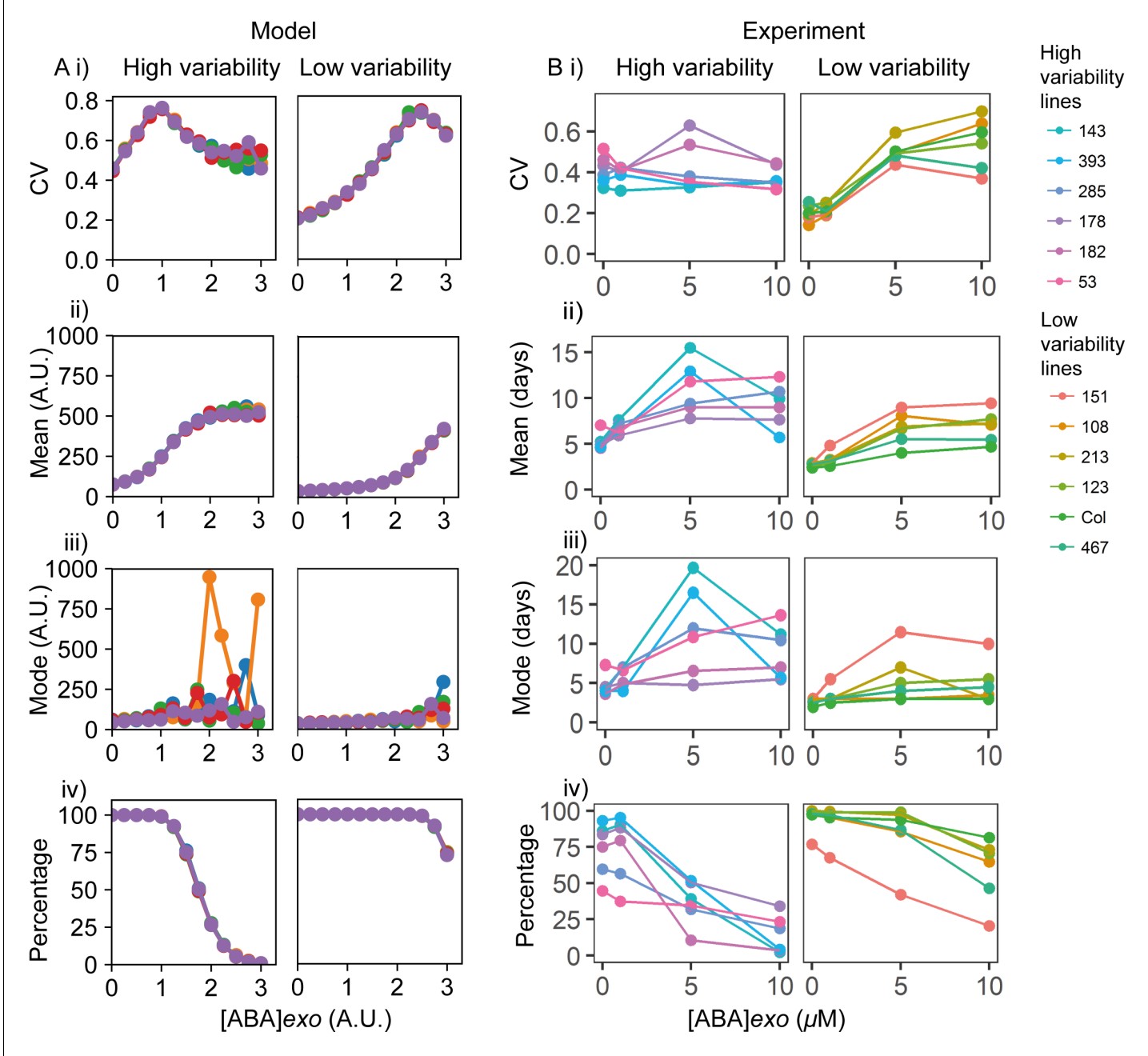

**Figure 6.** Exogenous addition of abscisic acid (ABA) to high and low variability lines. (**A**) Simulations of addition of increasing doses of exogenous ABA (x-axes), starting from a point in the parameter space that shows higher seed germination time variability (left) and lower variability (right) when no exogenous ABA is added. High variability in seed germination time is simulated with a lower value of the ABA threshold for I production ($\theta_{I,ABA}$) (i.e. higher ABA sensitivity) than low variability in seed germination time. Plots show the effects on the coefficient of variation (CV) (i), mean (ii), mode (iii) and percentage of seeds that germinated (iv) for the resulting germination time distributions. Each panel shows the result of five stochastic simulations for 4000 seeds, each plotted in a different colour. Parameter values for the high and low variability lines simulations are the same with the exception of the ABA threshold for I production ($\theta_{I,ABA}$ = 7 for the low variability lines and $\theta_{I,ABA}$ = 5.8 for the high variability lines). See Materials and methods for further simulation details and parameter values. (**B**) Experimental ABA dose response for six high variability MAGIC lines (left) and six low variability lines (five MAGIC lines plus Col-0) (right). (**B**) (i) shows mean CVs of individual lines for different exogenous ABA concentrations (means are of at least two independent experiments), (ii) as for (i) but for mean days to germination, (iii) mode days to germination and (iv) percentage germination. Treatments with '0' µM are vehicle control treatments. *Figure 6—figure supplement 1* shows exogenous addition of gibberellic acid (GA) in the model and experimentally to the high and low variability lines. *Figure 6—figure supplement 2* shows simulated germination time distributions for selected concentrations of exogenous ABA and GA. *Figure 6—figure supplement 3* shows the results of nullcline analysis in the presence of exogenous ABA

*Figure 6 continued on next page*

*Figure 6 continued*

and GA. *Figure 6—figure supplement 4* shows the effects of exogenous ABA and GA on germination time distributions for example high and low variability lines. *Figure 6—source data 1* contains source data for (B).

The online version of this article includes the following source data and figure supplement(s) for figure 6:

**Source data 1.** Figure6_ABAGAdosesGermSummaries.
**Figure supplement 1.** Exogenous addition of gibberellic acid (GA) to high and low variability lines.
**Figure supplement 2.** Simulated germination time distributions for a range of concentrations of exogenous abscisic acid (ABA) and gibberellic acid (GA).
**Figure supplement 3.** Results of nullcline analysis for abscisic acid (ABA) and gibberellic acid (GA) dose responses applied to the high variability parameter set.
**Figure supplement 4.** Effects of abscisic acid (ABA) and gibberellic acid (GA) on germination time distributions for example high and low variability lines.
**Figure supplement 4—source data 1.** Figure6_FigureSupplement4_ColM182_ABA_GA.

The high exogenous ABA can make the system become monostable, causing the non-germination solution to be the only one.

We next sought to experimentally test the model prediction that increasing exogenous ABA tends to increase variability in germination time. To do this we treated a number of high or low variability lines with a range of exogenous ABA doses and quantified germination at 1-day intervals. The low variability lines behaved similarly to each other under the addition of ABA (*Figure 6B*) and GA (*Figure 6—figure supplement 1B*), as did the high variability lines, irrespective of their specific QTL haplotypes on Chr3 and Chr5 (*Table 1*). This supports our use of ABA sensitivity as a general parameter for modulating germination time distribution in the model, rather than modelling specific lines and QTL haplotypes with different parameter sets. Consistent with the hypothesis that high and low variability lines differ in their ABA sensitivity, the percentage germination of high variability lines tended to be more sensitive to exogenous ABA compared to low variability lines (*Figure 6Biv*).

ABA treatments tended to increase the spread of germination time distributions, particularly for low variability lines. In the low variability lines, high ABA concentrations of 5 and 10 µM caused large increases in the CV of germination time, such that the distributions of germination times for low variability lines treated with ABA were similar to those of high variability lines in control conditions (*Figure 6Bi*, *Figure 6—figure supplement 4A*, compare Col-0, 5 and 10 µM [ABA]*exo* with M182, 0

**Table 1.** High and low variability lines used for abscisic acid (ABA) and gibberellic acid (GA) dose responses and their haplotypes at the Chr3 and Chr5 quantitative trait loci .

Haplotypes were classified according to their estimated effect on coefficient of variation (CV), as shown in *Figure 4—figure supplement 2*. Haplotype effect is classified as low/high when its average predicted effect is less than/higher than the mean haplotype effect, and the 95% confidence interval of the haplotype's effect does not overlap with mean haplotype effect. Haplotype effect is classified as medium when its 95% confidence interval overlaps with mean haplotype effect.

| Line | H/L variability | Chr3 haplotype | Chr3 haplotype effect on CV | Chr5 haplotype | Chr5 haplotype effect on CV |
|---|---|---|---|---|---|
| 143 | High | Can | Medium | Zu | High |
| 178 | High | Wu | High | Rsch | Medium |
| 182 | High | Edi | Medium | Hi | Medium |
| 285 | High | Ler | High | Zu | High |
| 393 | High | Ler | High | Kn | High |
| 53 | High | Sf | High | Rsch | Medium |
| 108 | Low | Tsu | Low | Rsch | Medium |
| 123 | Low | Bur | Medium | Wu | Medium |
| 151 | Low | Bur | Medium | Wu | Medium |
| 213 | Low | Wil | Low | Can | Medium |
| 467 | Low | Wil | Low | Edi | Low |
| Col-0 | Low | Col-0 | Low | Col-0 | Medium |

μM [ABA]*exo*). This was consistent with the prediction from the model that an initial increase in exogenous ABA concentration causes an increase in CV (*Figure 6Ai*, *Figure 6—figure supplement 2A*). Changes in mode were modest for low variability lines (*Figure 6Biii*), with a more obvious effect on percentage germination, which decreased at 10 μm (*Figure 6Biv*). These observations are compatible with the model (*Figure 6A*). The experimental observation that the moderate ABA dose of 5 μM caused larger increases in CV compared with the changes in the mode was also consistent with the model (*Figure 6Ai, iii*, *Figure 6—figure supplement 2A*, [ABA]*exo* = 0 compared to [*ABA*] *exo* = 1.5). For high variability lines, increasing concentrations of ABA increased mean and mode germination times, but had relatively little effect on CV (*Figure 6Bi-iii*, *Figure 6—figure supplement 4A*, M182). Some high variability lines show an increase in CV followed by a decrease, which is consistent with the model, and some lines show a slight decrease (*Figure 6Bi*). It is possible that these lines occupy different positions in parameter space due to variation at loci (and therefore components of the ABA-GA network) that are not accounted for in our simulations of these lines. Consistent with a less striking effect of exogenous ABA on the CV of high variability lines compared to low variability lines, in the model, the fold changes in CV upon ABA addition for high variability lines were smaller than those predicted for low variability lines (*Figure 6Ai*).

In contrast to ABA treatments, addition of exogenous GA to high and low variability lines in the simulations led to a reduced mode and less variable germination times (*Figure 6—figure supplement 1Ai, iii*, *Figure 6—figure supplement 2B*). This is because addition of GA can destabilise and even destroy the high Integrator, high ABA and low GA non-germination steady state (*Figure 6—figure supplement 3B*), causing the model to become monostable. In this situation, there is a rapid decrease of the Integrator, leading to rapid and less variable germination times (see dynamics of monostable scenario in *Figure 5—figure supplement 1A–G*).

As predicted by the model, when added experimentally to high and low variability MAGIC lines, GA tended to decrease the level of variability in germination time, with the effect strongest for the high variability genotypes (*Figure 6—figure supplement 1B*, *Figure 6—figure supplement 4B*, M182). As expected from previous studies (*Bewley, 1997*; *Koornneef and Karssen, 1994*; *Ni and Bradford, 1993*), high GA addition also increased germination percentages and tended to decrease the mean germination time in high variability lines (*Figure 6—figure supplement 1B*). For low variability lines, GA had little effect on the variability (CV), percentage germination, mean or mode germination times (*Figure 6—figure supplement 1B*, *Figure 6—figure supplement 4B*, Col-0) as these lines germinated in a less variable manner with high percentage germination even in the absence of GA. Thus, for both ABA and GA, the overall effects of exogenous addition were qualitatively similar between the model and experiments.

We also sought to investigate the effects of altered levels of ABA or GA on germination time distributions using mutants. To test the effect of increased ABA concentration in the low variability background of the Col-0 accession, we used the *cyp707a1* and *cyp707a1 cyp707a2* mutants, which lack enzymes required for ABA catabolism (*Kushiro et al., 2004*; *Okamoto et al., 2006*), effectively decreasing the ABA degradation rate. Similar to the effect of exogenous addition of ABA on low variability lines, loss of function of the CYP707A1 enzyme caused the Col-0 germination time distribution to become more long-tailed, with a small shift in the mode days to germination of ~1 day (*Figure 7*). Loss of both enzymes severely inhibited germination time and caused a large increase in the mode days to germination, similar to the effect of higher concentrations of exogenous ABA, which in both cases is related to the germination time distribution becoming more uniform (*Figure 7*). The correlated effect we found on CV, mode and percentage germination due to a change in the ABA degradation rate is also consistent with the results from changing the ABA degradation parameter in the model (*Figure 5—figure supplement 4Ai-iii*).

To test genetically the effect of decreasing the GA concentration on the germination time distribution of Col-0, we used the *ga3ox1-3 ga3ox2-1* mutant, which lacks two enzymes involved in GA biosynthesis (*Mitchum et al., 2006*). This double mutant, which has reduced GA levels (*Mitchum et al., 2006*), showed an increased CV and, similar to the *cyp707a1-1 cyp707a2-1* mutant, had increased mode and decreased percentage germination (*Figure 7*). Together with the GA and ABA dose–response experiments, these findings support the model predictions regarding the effects of altering ABA and GA levels on CV, mode and percentage germination.

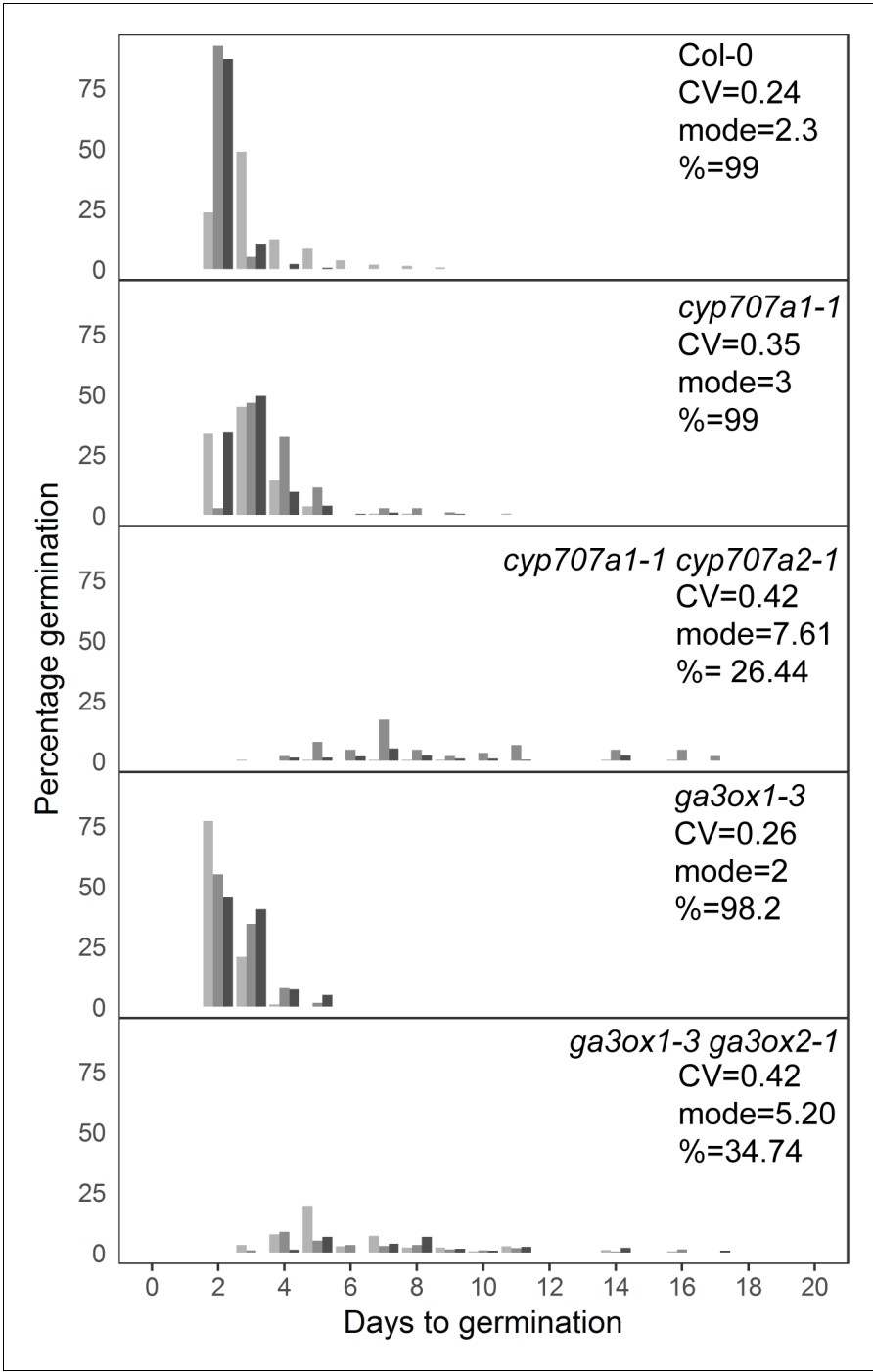

**Figure 7.** Mutants with altered levels of abscisic acid (ABA) and gibberellic acid (GA) have altered germination time variability. Distributions of germination times for indicated genotypes. *cyp707a1-1* and *cyp707a1-1 cyp707a2-1* mutants lack enzymes involved in ABA catabolism, while *ga3ox-3* and *ga3ox1-3 ga3ox2-1* mutants lack enzymes involved in GA biosynthesis. Plots show the percentage of all seeds that were sown, that germinated on a given day. Grey-coloured bars show the germination time distribution of seed batches from replicate mother plants. For Col-0 and each mutant, the mean coefficient of variation of germination times, mode days to germination and final percentage germination is shown (averaged across the replicate batches [n = 3]). Data are representative of at least two independent experiments for each genotype. Source data is provided in *Figure 7—source data 1*. The online version of this article includes the following source data for figure 7:

**Source data 1.** Figure7_MutantsGerm.

## Discussion

Variability in seed germination time is relevant for plant survival in the wild, where high variability may function as a bet-hedging strategy. The advantages of variability in the wild contrast to the situation in agriculture, where minimal levels of variability are desirable to promote crop uniformity needed for optimal harvests (*Finch-Savage and Bassel, 2016*; *Mitchell et al., 2017*). By describing detailed distributions of germination time for hundreds of Arabidopsis lines grown in a common environment, our work reveals that these distributions are genetically controlled since they can vary greatly between different lines and are reproducible for a given genotype. Using this natural variation, we identified two loci underlying variability and present evidence that candidate genes underlying the loci may alter germination time through influencing ABA sensitivity. Furthermore, modelling and experiments show that perturbation of the ABA-GA network can modulate variability in germination times in a predictable manner. These findings suggest that high or low variability could be specifically selected for both in the wild and in crop breeding programmes.

Previously it was shown that, in some Arabidopsis accessions, seeds matured on primary inflorescences had different germination behaviours to those that developed on branches (*Boyd et al., 2007*). It was also reported that developmental differences exist along the Arabidopsis inflorescence, and these developmental differences between flowers could, in principle, create differences in seed germination time between fruits (*Plackett et al., 2018*). Here we show that variability in germination time exists even for seeds from the same silique of Arabidopsis, suggesting that it can arise independently from developmental gradients in the plant. We also show that seeds from proximal and distal halves of siliques have similar germination behaviours, indicating that variability is likely not caused by gradients of regulatory molecules along the length of the fruit. Thus, our findings indicate that a mechanism exists to generate different behaviours amongst seeds that are as equivalent as possible.

We show that at least two genetic loci underlie germination time variability in the MAGIC population (*Kover et al., 2009*). The QTL at ~16 Mb on Chr3 overlaps with the *DELAY OF GERMINATION 6* (*DOG6*) locus (*Alonso-Blanco et al., 2003*; *Bentsink et al., 2010*), and is associated with CV, mean and mode germination time and percentage germination. There is published evidence from a GWAS study on seed dormancy that suggests that the *DOG6* locus corresponds to the *ANAC060* gene, which regulates ABA sensitivity (*Li et al., 2014*; *Hanzi, 2014*). When we tested the effect of the *anac060* mutation on germination time distributions and ABA sensitivity in our germination assays, we found a slight increase in ABA sensitivity and evidence for a small increase in the CV and mode of germination times compared to Col-0 in seeds sown 3 DAH. However, we did not observe a difference in germination time distributions between the mutant and Col-0 when seeds were stored for 30 DAH as was done in the QTL mapping. Thus, it may be the case that a gene other than *ANAC060* underlies the effect of our QTL, or if it does underlie it, then its effect may be stronger in other genetic backgrounds within the MAGIC lines, or there may be allele-specific effects on germination time distributions. We note that our results are in contrast with *Hanzi, 2014*, who showed a large effect of the same *anac060* T-DNA mutation on percentage germination.

The peak of the locus at ~18.6 Mb on Chr5 identified from our F2 bulked-segregant mapping approach overlaps with the *DOG1* gene (*Alonso-Blanco et al., 2003*; *Bentsink et al., 2006*), which feeds in to promote ABA signalling independently from the presence of ABA and thus increases sensitivity to ABA in germination assays (*Née et al., 2017a*; *Nishimura et al., 2018*). Consistent with this, and supporting *DOG1* being a candidate gene for the Chr5 locus, in our germination assays we observe a decrease in CV and mode of germination time distributions in the *dog1* mutant, in the Col-0 and No-0 accession backgrounds, which is associated with decreased ABA sensitivity. However, the peak of the Chr5 QTL identified in the MAGIC population lies at 19.8 Mb, equidistant between *DOG1* and a newly identified germination locus, *SET1*, which lies at around 21 Mb on chromosome 5 (*Footitt et al., 2020*). *SET1* was shown to influence the sensitivity of seeds buried in the soil to seasonal cycles of environmental conditions that influence germination time in the field (*Footitt et al., 2020*). It has been hypothesised that the *SET1* locus corresponds to the *ABA-HYPERSENSITIVE GERMINATION1* (*AHG1*) gene, which is itself inhibited by *DOG1* and acts to suppress ABA-imposed seed dormancy (*Fuchs et al., 2013*; *Née et al., 2017a*; *Nishimura et al., 2018*; *Footitt et al., 2020*). Consistent with this, we found a slight increase in CV and mode in the *ahg1* mutant background. Thus, it is possible that *SET1/AHG1* rather than *DOG1* underlies the Chr5

locus, or that both are relevant in the MAGIC lines, or that another gene underlies the effect of this locus.

In the QTL mapping, the Chr5 locus was only found to be significantly associated with the CV of germination time. However, besides their effect on CV, the mutant alleles of the two candidate genes we tested, *DOG1* and *AHG1*, also affected mode and percentage germination (*Bentsink et al., 2006*; *Née et al., 2017a*). The fact that we did not detect an association between this region of Chr5 and those two latter traits could be explained by differences between the naturally occurring alleles within the MAGIC population (which may predominantly affect CV) and the mutant alleles of these genes (which affect multiple traits). It could also be the case that the MAGIC lines' alleles affect multiple germination traits but that we did not have enough statistical power to detect this in the QTL mapping.

Further work will be needed to identify the causal variants underlying both of the QTL identified here and to determine the extent to which they have correlated or uncoupled effects on different germination traits. It will be important to test directly candidate genes through transforming candidate gene null mutants with corresponding alleles cloned from high and low variability lines (*Weigel, 2012*). These transgenic lines could then be used to understand how natural variation at the two QTL influences germination time distributions by using them to quantify the effects of the different accession alleles on CV, mode and percentage germination as well as on traits related to the ABA-GA network such as ABA and GA sensitivity. In future work, it would also be informative to repeat the QTL mapping for variability in seed germination time under alternative conditions for maternal growth, seed storage and sowing, thus testing the extent to which common loci are involved under different environmental conditions.

Our stochastic model of a simplified representation of the underlying ABA-GA network suggests that this bistable switch can generate variability in germination times and account for differences between MAGIC lines. We show that, when operating in the bistable region of parameter space after the rise in GA production that occurs upon sowing, the model tends to generate higher CVs and higher modal germination times compared with when operating in the monostable region. We also show that having a more stable non-germination steady state when operating within the bistable regime can lead to increased variability associated with increased mode and decreased percentage germination. Thus, by having both monostable and bistable behaviours, with different stabilities of the non-germination steady state, the model can generate a large range of 'phenotypes' in terms of CV, mode and percentage germination, accounting for the variation in these traits observed in MAGIC lines and natural accessions. We show that the main differences between high and low variability lines can be accounted for by assuming differences in the sensitivity to ABA, as suggested by the candidate genes for the QTL. We note that our parameter explorations reveal that other parameters, including those governing the rates of ABA, GA and Integrator production and degradation, as well as the GA-dependent degradation rate of the Integrator and the sensitivity of ABA production to Integrator levels, can all have similar effects to the ABA sensitivity parameter and could also account for the correlations between CV, mode and percentage germination in the MAGIC lines.

We found that other parameters caused a variety of other behaviours in terms of the relationship between the CV and mode of the germination time distributions. This is consistent with the observation of a complex relationship between CV and mode in the MAGIC lines and accessions, whereby these traits are weakly positively correlated between lines, but can be partially decoupled. Since it is likely that we have not detected all genetic loci that influence germination time distributions in our QTL mapping (the Chr3 and Chr5 QTL together only account for 23% of the variance in CV in the MAGIC lines), it is possible that undetected genetic variation in multiple components of the ABA-GA network contributes to the variation in germination time distributions that we observe and the complex relationship between CV, mode and percentage germination in the data.

Because germination in our model is driven by the rise in GA production that occurs upon sowing, it behaves as a time-dependent toggle switch (*Verd et al., 2014*) with stochastic fluctuations. In such time-dependent systems, steady states that exist in the beginning of the simulation can cease to exist or can change in their stability over time due to the changes in the time-dependent term (here, the GA production). The former occurs in what we refer to as the 'monostable scenario'. Here the stable non-germination steady state that exists before the rise in GA production can disappear in what is called a saddle-node bifurcation, causing seeds to transition directly to the germination steady state, resulting in rapid germination. In the bistable scenario, the non-germination stable

steady state remains after the rise in GA production, allowing seeds to exist in this state for some time. Stochastic fluctuations can then result in seeds being displaced sufficiently from this state so that they are then attracted to the stable steady state associated with germination. The more stable the non-germination state is, the less readily seeds are sufficiently displaced from it by stochastic fluctuations to cause transition to the germination steady state, meaning that the time required for all seeds to experience a large enough fluctuation to cause them to germinate becomes larger, causing more variable germination times. Thus, our work shows that a time-dependent toggle switch (*Verd et al., 2014*) with stochastic fluctuations can be used as a framework to understand variable germination behaviour.

In future work, this framework can be used to integrate additional components of the ABA-GA network. Recent work has shown that feedback acting upon ABA biosynthesis and catabolism can modulate variability in ABA concentrations independent of the mean concentration (*Johnston and Bassel, 2018*). In the future, it will be possible to extend our initial ABA-GA model to evaluate the effects of different feedback motifs acting on the two hormones. It is likely that with a more detailed implementation of the network the model would have an increased ability to ascribe different germination traits to specific parameters, allowing us to make further mechanistic hypotheses about the effects of particular loci. With such a model, it might also be possible to capture the rare bimodal distributions that we observe in some MAGIC lines.

One limitation of our model is that we represent a seed as a single compartment in which we simulate the effects of stochastic molecular interactions. In fact, the decision to germinate is likely made by groups of cells in the embryonic root (*Mitchell et al., 2020*; *Topham et al., 2017*), and thus it is unclear how the noise arising in individual cellular compartments would influence this multicellular decision. One possibility is that noise would be averaged out across the cells, reducing the level of noise at the multicellular level. This appears to be the case during vernalisation, where polycomb-based epigenetic silencing of the floral repressor *FLOWERING LOCUS C* (*FLC*) occurs in response to cold (*Angel et al., 2011*; *Song et al., 2012*). In a model of this system (*Angel et al., 2015*), the silencing of each *FLC* locus in each cell is proposed to be a probabilistic event, with the likelihood of silencing increasing with the duration of cold treatment. This generates heterogeneity in *FLC* expression at the individual cell level, which is then averaged out at the level of downstream processes, resulting in the plant flowering at a time that accurately reflects the duration of cold exposure.

On the other hand, noise in gene expression and phenotypic outputs has been reported for multiple pathways in plants (*Jimenez-Gomez et al., 2011*; *Joseph et al., 2015*), and recent work quantifying genome-wide transcriptional variability among individual seedlings in the same environment shows that, for many genes, there is a large degree of between-plant variability in transcript levels (*Cortijo et al., 2019*), suggesting that variability is not always averaged out to give uniformity of multicellular individuals. One possibility is that for some genetic networks high levels of variability in gene expression arise because the properties of the network mean that stochasticity is amplified. Interestingly, the transcriptomic study of *Cortijo et al., 2019* showed that genes that were highly variable in their expression between seedlings were enriched for those involved in the response to ABA, supporting the idea that the bistable switch governing ABA levels amplifies stochasticity to generate variability in gene expression and phenotypes. In further support of a potential role of the ABA-GA bistable switch in regulating phenotypic variability beyond germination, a recent study implicated redundantly acting GA receptors of tomato in maintaining phenotypic stability of plant growth (i.e. low variability) under ambient conditions (*Illouz-Eliaz et al., 2019*).

Our work provides the foundation for future functional and mechanistic work on phenotypic variability using germination time as a model system. Alternative functions beyond bet-hedging can be explored, including the possible function of distributed germination times in avoiding seedling competition, or that the variable germination could contribute to the seed bank, which acts as a persistent source of genetic diversity that may be advantageous for long-term species survival in variable environments (*Templeton and Levin, 1979*). Further mechanistic insights would be greatly increased by the development of new non-destructive techniques to track gene expression in seeds and correlate it with their germination time, which is challenging because of technical difficulties in imaging through the seed coat. However, destructive techniques have already revealed significant differences in the level of germination regulators in individual seeds of tomatoes (*Still et al., 1997*; *Still and Bradford, 1997*) and allowed measurement of ABA in individual seeds of a lowly variable Arabidopsis line (Col-0) (*Kanno et al., 2010*). It would be interesting to extend studies such as these to the

lines identified here with high and low variability in germination times. In general, our work emphasises that the level of variability in germination times is an important trait that does not always correlate with the most common measure of germination behaviour, which is percentage germination. Thus, to fully understand the roles of germination regulators, future work should involve characterising their effects on the CV of germination time distributions as well as on traits such as percentage germination and average germination time.

Multiple other aspects of Arabidopsis development have been shown to display phenotypic variability, including both whole plant phenotypes such as growth rate and molecular-level phenotypes such as transcript abundance (*Hall et al., 2007*; *Cortijo et al., 2019*; *Jimenez-Gomez et al., 2011*; *Joseph et al., 2015*; *Shen et al., 2012*). Similar to our findings, a common feature across studies and traits is that multiple QTL tend to affect variability of a trait, and while some of them also affect the average trait value, others do not. By establishing variability in germination time as a robust trait that varies between natural accessions of Arabidopsis, our study reveals that germination time is a plausible system for studying the genetics of inter-individual variability.

# Materials and methods

### Key resources table

| Reagent type (species) or resource | Designation | Source or reference | Identifiers | Additional information |
|---|---|---|---|---|
| Biological samples (*Arabidopsis thaliana*) | MAGIC lines | NASC | NASC ID: N782242 | PMID:19593375 |
| Biological samples (*Arabidopsis thaliana*) | MAGIC parental accessions | NASC | IDs of individual parental accessions: http://arabidopsis.info/CollectionInfo?id=112 | PMID:19593375 |
| Biological samples (*Arabidopsis thaliana*) | Spanish accessions | NASC | | PMID:26991665 |
| Biological sample (*Arabidopsis thaliana*) | *cyp707a1-1* | PMID:16543410 | | Eiji Nambara, University of Toronto |
| Biological sample (*Arabidopsis thaliana*) | *cyp707a1-1 cyp707a2-1* | PMID:16543410 | | Eiji Nambara, University of Toronto |
| Biological sample (*Arabidopsis thaliana*) | *ga3ox1-3* | NASC | NASC ID: N6943 | PMID:16460513 |
| Biological sample (*Arabidopsis thaliana*) | *ga3ox1-3 ga3ox2-1* | NASC | NASC ID: N6944 | PMID:16460513 |
| Biological sample (*Arabidopsis thaliana*) | *dog1-3* (Col-0) (SALK 000867) | NASC | NASC ID: N500867 | PMID:17065317 |
| Biological sample (*Arabidopsis thaliana*) | *anac060* (SALK 012554C) | NASC | NASC ID: N665285 | PMID:24625790 |
| Biological sample (*Arabidopsis thaliana*) | *ahg1-5* | PMID:28706187 | | Guillaume Néé, University of Münster |
| Biological sample (*Arabidopsis thaliana*) | *dog1* (No-0) (*dog1* mutant in No-0 background) | RIKEN | BRC number: pst21966 Line number: 15-3980-1 | |
| Sequence-based reagent | DOG1N_F | This paper | PCR primers | GAAATCCGCTCCTTGTACCG See *Supplementary file 4* |
| Sequence-based reagent | DOG1N_R | This paper | PCR primers | GCATCCCTGAGCTCAAACAA See *Supplementary file 4* |
| Sequence-based reagent | Ds5-2a | PMID:14996221 | PCR primers | TCCGTTCCGTTTTCGTTTTTTAC See *Supplementary file 4* |
| Sequence-based reagent | DOG1-3 F | This paper | PCR primers | TTCCAGGAACGTTGTCGTATC See *Supplementary file 4* |
| Sequence-based reagent | DOG1-3 R | This paper | PCR primers | AGTTTGTGACCCACACAAAGC See *Supplementary file 4* |

*Continued on next page*

*Continued*

| Reagent type (species) or resource | Designation | Source or reference | Identifiers | Additional information |
|---|---|---|---|---|
| Sequence-based reagent | LBb1.3 | http://signal.salk.edu/tdnaprimers.2.html | PCR primers | ATTTTGCCGATTTCGAAC See *Supplementary file 4* |
| Sequence-based reagent | ANAC060 F | This paper | PCR primers | TGGACTCTGTTTGAAGCCTTG See *Supplementary file 4* |
| Sequence-based reagent | ANAC060 R | This paper | PCR primers | TATGCCTGTCCTGATTTGCTC See *Supplementary file 4* |
| Sequence-based reagent | AHG1 LP | PMID:28706187 | PCR primers | ACCGACACGTGTTCTGTCTTC See *Supplementary file 4* |
| Sequence-based reagent | AHG1 RP | PMID:28706187 | PCR primers | CTAAAACTCGACCACCAGCTG See *Supplementary file 4* |
| Chemical compound, drug | Gibberellin A$_4$ | Sigma Aldrich | G7276 | |
| Chemical compound, drug | Abscisic acid | Sigma Aldrich | A1049 | |
| Commercial assay, kit | NEB Next Ultra DNA Library Prep Kit | New England BioLabs | E7370L | |
| Software, algorithm | Organism | PMID:15961462 | | https://gitlab.com/slcu/teamHJ/Organism |
| Software, algorithm | R | R Foundation for Statistical Computing | | https://www.R-project.org/ |
| Software, algorithm | Python | Python Software Foundation | | Version: Python 2.7 https://www.python.org/download/releases/2.7/ |
| Software, algorithm | Data analysis and modelling scripts | This paper | | https://gitlab.com/slcu/teamJL/abley_formosa_etal_2020 ; *Abley et al., 2021* copy archived at swh:1:rev:0a97b841e58b128c174d93fc759b28f1df2966a2 |

## Plant materials

MAGIC lines (*Kover et al., 2009*) and accessions (*Vidigal et al., 2016*) were obtained from the Nottingham Arabidopsis Stock Centre (NASC). The *cyp707a1-1* and *cyp707a1-1 cyp707a2-1* mutants are as described in *Okamoto et al., 2006* and were kindly provided by Eiji Nambara. *ga3ox1-3* and the *ga3ox1-3 ga3ox2-1* mutants are as described in *Mitchum et al., 2006* and were obtained from NASC. The *dog1-3* mutant in the Col-0 background is SALK 000867 (as described in *Bentsink et al., 2006*), and the *anac060* mutant is SALK 012554C (as described in *Li et al., 2014*). The *ahg1-5* mutant is as described in *Née et al., 2017a* and was kindly provided by Guillaume Neé. *dog1-3*, *anac060* and *ahg1-5* were all confirmed to be homozygous by PCR based genotyping, with primers detailed in *Supplementary file 4*. The *dog1* mutant in the No-0 background was obtained as a segregating Ds transposon tagged line from RIKEN, BRC number: pst21966, line number: 15-3980-1 (*Kuromori et al., 2004*). We obtained homozygous mutants by PCR-based genotyping with primers detailed in *Supplementary file 4*. We also obtained homozygous wild-type lines from the pst21966 segregating population and used them as controls in the germination assay (labelled as No-0 in *Figure 4—figure supplement 5*).

To generate seed for assaying germination time distributions, plants were grown in batches of 40 lines in P40 trays with F2 soil treated with Intercept 70WG (both Levington, http://www.scottsprofessional.co.uk). In total, 345 MAGIC lines and 29 accessions were phenotyped. The parental plants used for seed collection were sown in a staggered manner across 13 batches. We checked that the sowing batch of the parental plants was not a major contributor to the variation seen between lines (~6% of the total phenotypic variance for CV could be attributed to the sowing batch, while 52% was due to the genotype of the line).

Plants for seed collection were grown in Conviron growth chambers with 16 hr of light (170 μM/m$^2$/s) and 8 hr of dark, with a day time temperature of 21°C and night-time temperature of 17°C at

65% relative humidity. These are standard conditions for Arabidopsis growth and similar or the same as those used for seed harvest in a number of studies (*Donohue et al., 2005*; *Finch-Savage et al., 2007*; *Linder, 2014*; *Springthorpe and Penfield, 2015*).

Some of the accessions required vernalisation to flower. For these lines, after 10 days of growth in the standard conditions described above, the plants were transferred to a Conviron growth chamber with 8 hr of light (15 µM/m²/s) and 16 hr of dark, with a constant temperature of 5°C at 90% relative humidity. For the MAGIC parental accessions that were vernalised, the plants were kept in the cold for a period of 8 weeks. For the Spanish accessions, different lengths of vernalisation period were used, as described in *Vidigal et al., 2016*. Details of which accessions were vernalised, and the period of vernalisation for the Spanish accessions, are provided as Supplementary files *Supplementary file 2* and *Supplementary file 3*. To ensure that all plants used for seed harvest for QTL mapping were treated in an identical way, none of the MAGIC lines were vernalised; thus, only MAGIC lines that flowered without vernalisation were used in our study.

In a given sowing, genotypes were distributed across all trays (in random positions) and the trays were rotated approximately every three days to make sure that the parent plants were exposed to as similar microenvironmental conditions as possible. Six replicate plants of each line were grown. Each plant was bagged as soon as its first siliques started to ripen. Plants were watered until most (~95%) of the siliques had ripened, and then watering was stopped and plants were left in the growth chamber for 7 days to dry (*Huang et al., 2014*). Seeds obtained from these plants were then stored for approximately 30 days before sowing (e.g. *Linder, 2014*) in a dark chamber kept at 15°C and 15% relative humidity. To check the quality of seed collected and stored for ~30 days in these conditions, we performed stratification experiments for a subset of MAGIC lines (32 lines, including the most highly variable lines) by putting imbibed seeds at 4°C in the dark for 4 days prior to sowing. All but three lines had >90% germination after stratification, and the three lines that germinated poorly after stratification showed >97% germination when sown on plates containing 10 µM gibberellin $A_4$ (Sigma Aldrich, G7276).

Prior to sowing, seeds were sterilised for 4 min with 2.5% bleach, followed by one rinse with 70% ethanol for 1 min and then washed four times with sterile water.

After sterilising, seeds were suspended into 0.1% agar and pipetted on to an empty Petri dish, with even spacing between seeds. 0.9% agar was melted, cooled to 35°C, then poured on top of the seeds (25 ml per round Petri dish) and allowed to dry. This method of sowing seeds below agar makes scoring seeds over long time periods easier and helps to maintain a more constant environment than sowing on top of agar (where condensation forms) or on filter paper where it is difficult to maintain constant moisture levels. We checked the germination time distributions for 20 lines that were sown above agar (by pipetting seeds on top of solidified and cooled agar) or below agar, as described above, and found that we obtained similar CVs of germination time for both methods. Plates were sealed with micropore tape and put into a tissue culture room with 16 hr of light (85 µM/m²/s) and 8 hr of dark, a day time temperature of 20.5°C a night-time temperature of 18.5°C and 50% relative humidity. Each Petri dish contained approximately 150 seeds. Seed germination was scored daily, using a dissecting microscope to detect radicle protrusion, and plates were checked until at least 2 weeks after the last germination event was observed (with the exception of ABA and GA dose–response experiments, which were scored until 19 or 20 days after sowing, see ABA and GA dose–response methods section). The germination time data is provided in data files *Supplementary file 1* , *Supplementary file 2* and *Supplementary file 3* . To calculate germination statistics, the data were filtered to exclude plates where less than 10 seeds germinated. We reasoned that a minimum number of seeds was needed to reliably estimate the CV. This filtering meant that 4 MAGIC lines out of the 345 that we phenotyped were excluded completely from further analysis. For 24 MAGIC lines, one or more replicate seed batches were excluded from further analysis. Following filtering, for 91% of MAGIC lines, at least three replicate seed batches (each collected from a different parent plant) were used for each genotype. The seed batches were sown separately, with one Petri dish for each of the three batches. For 9% of lines, only one or two replicates were used (25 lines had two batches, 6 had only one batch).

For 32 MAGIC lines, the whole experiment was repeated, with parental plants for seed harvest from a new independent sowing. The germination time distributions of MAGIC lines and natural accessions were all determined on agar plates as described above. The Col-0 × No-0 F2 experiment was phenotyped on soil in the same conditions as those described above for growing plants for

seed harvest. Transparent lids were kept on the trays of soil, and newly germinated seedlings were removed and counted every day. A seed was considered to have germinated when its cotyledons had visibly emerged.

## QTL mapping by bulked-segregant analysis

Col-0 × No-0 F2 seeds were sown on soil as described above. Seedlings that germinated early (day 4, early pool, E1) or in two late pools (late one pool, L1: days 31–39 and late two pool, L2: days 43–60) were moved into separate P40 trays and grown until flowering. The primary apices of 152 plants from E1, 321 from L1 and 213 plants from L2 were collected onto dry ice shortly after bolting and stored at −80°C. The apices from each pool were combined and ground together in liquid nitrogen, and then genomic DNA was extracted using a CTAB method (*Glazebrook and Weigel, 2002*). Genomic DNA library preparation and sequencing was carried out by Novogene (UK) Company Limited using NEB Next Ultra DNA Library Prep Kit (cat no. E7370L). The libraries were sequenced on an Illumina NovaSeq 6000 machine with 300 cycles (150 bp paired-end reads).

All of the bioinformatics processing steps and options used are detailed in the scripts provided with this paper (see data availability), so we only provide a brief summary here. We used *FastQC v0.11.3* (*Andrews, 2020*) for checking read quality. Quality filtering was performed using *cutadapt 1.16* (*Martin, 2011*) to remove Illumina adapters from the reads, remove reads with ambiguous base calls and trim reads if the base quality dropped below a phred score of 20, keeping only those reads with at least 50 bp after trimming. Over 99% of bases were retained after filtering. The filtered reads were aligned to the Arabidopsis reference genome (TAIR10 version) using *bwa mem 0.7.12* (*Li and Durbin, 2009*) with default options, except we used the '-M' option to mark short split alignments as secondary. Potential PCR duplicates were removed using *Picard MarkDuplicates 2.18.1* (~20% of the reads were marked as duplicates). We performed realignment around indels using the '*RealignerTargetCreator*' tool from *GATK 3.4–46* (*McKenna et al., 2010*). Finally, we obtained allele counts at variant sites using *freebayes v1.2.0* (*Garrison and Marth, 2012*) using the '–pooled-continuous' mode, and restricting calls to sites with a depth of coverage between 10 and 400 including a minimum Phred base quality score and read mapping quality score of 20, and a minimum count of 2 and minimum frequency of 1% for the non-reference allele. We did not include indels in the analysis. To assess the presence of a QTL from these data, we used both the G' statistic of *Magwene et al., 2011* and the simulation-based approach of *Takagi et al., 2013* to compare the three pools with each other. Both of these methods are implemented in the *R/QTLseqr v0.7.5.2* package (*Mansfeld and Grumet, 2018*) and gave similar results, so we report only the latter.

## QTL mapping in MAGIC lines

QTL mapping in the MAGIC lines was performed using the *happy.hbrem R* package (*Kover et al., 2009*) and our custom package *MagicHelpR v0.1* (available at https://github.com/tavareshugo/MagicHelpR). In summary, for each of the 1254 available markers, the probability of ancestry of an individual's genotype at that marker was inferred using the function '*happy*' from the *R/happy.hbrem* package (*Mott, 2015*). For each marker, an N × 19 matrix is obtained with the probabilities that the N individuals inherited that piece of genome from each of the 19 founder accessions of the MAGIC population. This matrix was then used to fit a linear model, regressing the trait of interest onto this probability matrix. This type of model was used for each trait analysed. In all cases, significance was assessed using an F-test to compare the full model to a reduced model that excluded the genotype matrix, and we report the −log10(p-value) of this test. We used a genome-wide significance threshold of −log10(p-value) = 3.5, which is an approximate threshold at $\alpha = 0.05$ based on simulations (*Kover et al., 2009*). Significance of candidate QTL was also confirmed from a permutation-based empirical p-value based on 1000 phenotype permutations. The variance explained by candidate QTL markers was obtained from the coefficient of determination ($R^2$) of the linear model.

The founder accession's effect at each candidate QTL was estimated using the method in *Kover et al., 2009* and adapted from the *R* function '*imputed.one.way.anova*' in the *magic.R* script available at http://mtweb.cs.ucl.ac.uk/mus/www/magic/ (last accessed May 2020). In summary, each MAGIC line was assigned to a single-founder accession based on its ancestry probabilities at that marker. MAGIC lines are then grouped by founder accession, and the trait's average is calculated for the 19 accessions. This procedure was repeated 500 times to produce an average estimate and

associated 95% confidence interval (taken as the 0.025 and 0.975 quantiles of the phenotype distributions thus obtained).

All trait data were rank-transformed to achieve normality and constant variance of the residuals in the QTL model. However, our results were robust to data transformation.

## ABA and GA dose–response experiments

For *Figure 6* and *Figure 6—figure supplement 1*, dose–response experiments were performed on six high variability MAGIC lines (M143, M393, M285, M178, M182, M53) and five low variability MAGIC lines (M151, M108, M123, M213, M467), plus Col-0. Seed batches were pools of seed from three parent plants of each genotype. Dose responses were performed at least twice using independently collected seed batches for each MAGIC line used, except for MAGIC lines 467 and 151, which have one replicate in the GA dose response experiment.

Seeds were obtained, sowed and grown as described above for phenotyping, except that the indicated concentrations of gibberellin $A_4$ (Sigma Aldrich, G7276) or abscisic acid (Sigma Aldrich, A1049) were added to the 0.9% agar medium used for germination assays. The $GA_4$ stock was made using ethanol, and the ABA stock using methanol and respective vehicle control treatments were used. Germination was scored as radicle emergence until 20 days after sowing.

The ABA dose responses on candidate gene mutants in *Figure 4—figure supplement 5* were performed as described above, and germination was scored until 19 days after sowing. For *dog1-3* and Col-0 seeds sown 5 DAH in the absence of ABA, five independent experiments were performed, each with three separate seed batches from different parent plants. For *anac060* and Col-0 seeds sown 3 DAH in the absence of ABA, four independent experiments were performed, each with three separate seed batches from different parent plants. For all other mutant versus wild-type comparisons and the ABA dose responses, one experiment was performed with three separate seed batches from different parent plants.

## Minimal mathematical model for seed germination

We developed a model to capture the relationships between the hormones ABA and GA and key transcriptional regulators that act as inhibitors of germination. ABA and GA are known to have opposing effects on the transcription, protein levels or protein activity of the transcriptional regulators DELLAs, ABI4 and ABI5 (*Ariizumi et al., 2008*; *Liu et al., 2016*; *Piskurewicz et al., 2008*; *Shu et al., 2016a*; *Tyler et al., 2004*). Here we represent these germinator inhibitors as one factor, called Integrator, the production of which is promoted by ABA and the degradation of which is promoted by GA.

The germination inhibitors are known to feedback to influence GA and ABA levels through effects on their biosynthesis or catabolism (*Ko et al., 2006*; *Oh et al., 2007*; *Piskurewicz et al., 2008*; *Shu et al., 2016a*, *Shu et al., 2013*). For example, DELLAs promote the levels of expression of *XERICO*, which promotes ABA biosynthesis (*Ko et al., 2006*; *Zentella et al., 2007*). We capture this in the model by assuming that Integrator promotes ABA biosynthesis, creating a positive feedback loop between ABA and Integrator. Since GA inhibits the Integrator, GA ends up effectively inhibiting ABA levels through the integrator.

With regards to the feedback between the germination inhibitors (represented by the Integrator) and GA, the literature is less clear about the nature of the interaction. ABI4 appears to negatively regulate GA levels (*Shu et al., 2013*), supporting a double-negative (i.e. positive) feedback loop between GA levels and Integrator. However, there are mixed reports about the relationship between DELLAs and GA during germination, with studies suggesting both inhibition (*Oh et al., 2007*) and promotion (*Topham et al., 2017*; *Zentella et al., 2007*). As has previously been suggested (*Yamaguchi and Kamiya, 2000*), we assume that, on balance, the net relationship between the germination inhibitors and GA is negative during germination, creating a mutual inhibition between the inhibitors and GA levels. This may contribute to the large increases in GA levels that occur following sowing. With this set of interactions, since ABA increases the levels of Integrator, it effectively inhibits GA. Thus, the model captures the mutual inhibition between ABA and GA. Overall, the model exhibits a mutual inhibition and mutual activation circuit coupled by Integrator (see *Figure 5A*), constituting a double-positive feedback.

The deterministic model for ABA ([ABA]), GA ([GA]) and the Integrator ([I]) is described by the following equations:

$$\frac{d[ABA]}{dt} = \beta_{ABA} + f_{ABA}([I]) - v_{ABA}[ABA] \tag{1}$$

$$\frac{d[GA]}{dt} = \beta_{GA} + \beta_{GA,Z}[Z] + g_{GA}([I]) - v_{GA}[GA] \tag{2}$$

$$\frac{d[I]}{dt} = \beta_I + f_I([ABA]) - (v_I + f_I([GA]))[I], \tag{3}$$

where $\beta_X$ and $v_X$ are constitutive production and degradation rates for each $X$ variable, respectively, and $\beta_{GA,Z}$ is a coefficient for another production term for GA, coming from a first-order reaction to simulate the sowing-induced increase in GA production (see below). $f_X(y)$ and $g_X(y)$ correspond to Hill increasing and decreasing regulatory functions acting on variable $X$, defined as $f_X(y) = \frac{C_{X,Y} y^h}{\theta_{X,Y}^h + y^h}$ and $g_X(y) = \frac{C_{X,Y}}{1 + \frac{y^h}{\theta_{X,Y}^h}}$, respectively, where $C_{X,Y}$ and $\theta_{X,Y}$ are parameters in the function dependent on variable $Y$ acting on variable $X$, and $h$ is the exponent in these functions. For simplicity, we set all exponents to the same value. We refer to $\theta_{X,Y}$ parameters as regulatory thresholds, representing the $Y$ concentration value at which the regulatory function is half of its maximal value. Note that the inverse of $\theta_{X,Y}$ parameters can be understood as sensitivities to $Y$ acting on $X$; high $\theta_{X,Y}$ values will generally require high $Y$ quantities to affect $X$ dynamics through the regulatory function, meaning low sensitivity of $X$ to $Y$.

We simulate a sowing-induced increase in GA production rates by adding an extra factor, $[Z]$, which follows the dynamics governed by *Equation 4* and feeds into *Equation 2*:

$$\frac{d[Z]}{dt} = \beta_Z - v_Z[Z]. \tag{4}$$

We focus on those parameters leading to either monostability or bistability, typically showing a low GA – high ABA – high Integrator stable state and a high GA – low ABA – low Integrator stable state. To capture the inhibitory effect of the DELLAs, ABI4 and ABI5 (represented by Integrator) on germination, we assume that in each seed the Integrator level must drop below a threshold for germination to occur. If the system switches to the low GA – high ABA – low Integrator state, and this Integrator state is below a certain threshold, then germination occurs.

Our model can be understood as a time-dependent switch (*Verd et al., 2014*) with stochastic fluctuations, and variability in timing – in this case, germination time – is captured when crossing a concentration threshold (*Ghusinga et al., 2017*). Note that this circuit can also lead to tristability, but, for simplicity, we do not explore this model feature in detail.

In simulations of exogenous ABA or GA application, the Integrator equation follows the dynamics

$$\frac{d[I]}{dt} = \beta_I + f_I([ABA] + [ABA]_{exo}) - (v_I + f_I([GA] + [GA]_{exo}))[I], \tag{5}$$

where $[ABA]_{exo}$ and $[GA]_{exo}$ are constant variables representing the concentrations of exogenous ABA and GA.

To take into account the intrinsic fluctuations of the network, we simulated the stochastic chemical Langevin equations (*Gillespie, 2000*; *Adalsteinsson et al., 2004*) of the model *Equations 1–4*, which read

$$\frac{d[ABA]}{dt} = \beta_{ABA} + f_{ABA}([I]) - v_{ABA}[ABA] + \sqrt{\frac{1}{2V}(\beta_{ABA} + f_{ABA}([I]) + v_{ABA}[ABA])}\,\eta_{ABA}(t) \tag{6}$$

$$\frac{d[GA]}{dt} = \beta_{GA} + \beta_{GA,Z}[Z] + g_{GA}([I]) - v_{GA}[GA] + \sqrt{\frac{1}{2V}(\beta_{GA} + \beta_{GA,Z}[Z] + g_{GA}([I]) + v_{GA}[GA])}\,\eta_{GA}(t) \tag{7}$$

$$\frac{d[I]}{dt} = \beta_I + f_I([ABA]) - (v_I + f_I([GA]))[I] + \sqrt{\frac{1}{2V}(\beta_I + f_I([ABA]) + (v_I + g_I([GA]))[I])}\,\eta_I(t) \tag{8}$$

$$\frac{d[Z]}{dt} = \beta_Z - v_Z[Z] + \sqrt{\frac{1}{2V}(\beta_Z + v_Z[Z])}\,\eta_Z(t), \tag{9}$$

where $V$ is an effective volume of the modelled system, which determines the strength of the stochastic term; $\eta_X$ is a Gaussian random number with zero mean that fulfils $<\eta_X(t)\eta_Y(t')> = \delta(t-t')\delta_{X,Y}$; $\delta_{X,Y}$ is the Kronecker delta, where $X$ and $Y$ refer to concentration variables and $\delta(t-t')$ is the Dirac delta, where $t$ and $t'$ are two arbitrary time points. We will refer to noise intensity as the inverse of the $V$ parameter, given that the stochastic terms diminish with the increase of $V$. Note that all stochastic equations recover the deterministic limit when $V$ parameter goes to infinity, as expected for the standard chemical Langevin equation (*Gillespie, 2000*).

The stochastic version for *Equation 5* for modelling the application of exogenous ABA and GA reads

$$\frac{d[I]}{dt} = \beta_I + f_I([ABA] + [ABA]_{exo}) - (v_I + f_I([GA] + [GA]_{exo}))[I]$$

$$+ \sqrt{\frac{1}{2V}(\beta_I + f_I([ABA] + [ABA]_{exo}) + (v_I + f_I([GA] + [GA]_{exo}))[I])}\,\eta_I(t). \tag{10}$$

This bistable switch model is reminiscent of the bistable switch model proposed by *Topham et al., 2017*, although the mutual inhibition has been implemented differently, and we considered stochastic fluctuations.

Initial conditions were set at the fixed point of the deterministic model that exhibited the highest Integrator value before the sowing-induced increase in GA production (i.e. the highest root solution for the Integrator when $\beta_{GA,Z}=0$). When exogenous ABA or GA were applied, we assumed that seeds were in the same initial state as they would have been in the absence of exogenous hormone treatments. Numerical integration of the chemical Langevin equations with the îto interpretation was performed with the Heun algorithm (*Carrillo et al., 2003*) with an absorptive barrier at 0 to prevent negative concentration values. After each integration step, seeds were tagged as germinated if their Integrator concentration was below the germination threshold. The integration time step was set at $dt$ = 0.1. All simulations were stopped at time 1000.

Fixed points of the deterministic dynamics were computed by finding the solutions to the nullclines for the deterministic model *Equations 1–4*, that is, $d[ABA]/dt=d[GA]/dt=d[I]/dt=d[Z]/dt = 0$, and then by substituting all the variables into the Integrator equation. The algebraic equation to solve reads

$$[I]_0 = \frac{\beta_I + f_I([ABA]_0)}{v_I + f_I([GA]_0)}, \tag{11}$$

with

$$[ABA]_0 = \frac{\beta_{ABA} + f_{ABA}([I]_0)}{v_{ABA}} \tag{12}$$

$$[GA]_0 = \frac{\beta_{GA} + \beta_{GA,Z}[Z]_0 + g_{GA}([I]_0)}{v_{GA}} \tag{13}$$

$$[Z]_0 = \frac{\beta_Z}{v_Z}, \tag{14}$$

where $[ABA]_0$, $[GA]_0$, $[I]_0$ and $[Z]_0$ are the steady state solutions of the different variables. For finding the fixed points in the cases of exogenous application of GA and ABA, an equivalent procedure was performed with *Equations 1, 2, 4 and 5*.

To find all the solutions at each particular parameter set, we first used the bisection method throughout logarithmically spaced intervals for the Integrator variable to find approximate solutions, and then used the opt.brentq scipy function in Python to find the exact solutions. We also represented the left- and right-hand side of *Equation 11* or an equivalent equation for the exogenous application of GA and ABA to graphically see the solutions of the system (see *Figure 5—figure supplement 1* and *Figure 6—figure supplement 3*). In these plots, we represented the deterministic stability by analysing the sign of dI/dt at the vicinity of the solutions (*Strogatz, 2015*). Upon the variation of a certain parameter value or the application of exogenous ABA or GA, a given stable fixed point will approach to (or get further from) the separatrix, i.e. the hyperplane in the solution space separating the basins of attraction of the stable fixed points (*Strogatz, 2015*), which contains an unstable fixed point in our case. When studying the stochastic system, we will assume that a stable fixed point will lose stability when it approaches the unstable fixed point, given that this most likely will facilitate the stochastic switching to the other stable fixed point; conversely, a stable fixed point will gain stability when it gets further from the unstable fixed point. This assumption is consistent with the outcome of our simulations (*Figure 5—figure supplement 1H–U*, *Figure 6—figure supplement 2A*, *Figure 6—figure supplement 3A*).

Parameter values for the simulations and the theoretical plots were set as described in *Table 2*.

To better understand the dynamics across several parameter ranges, we performed simulations varying two parameters at the same time. The resolution of the parameter exploration was of 4–5 parameter values per order of magnitude, logarithmically spaced. In this parameter space exploration, we also studied the different regions of the parameter space that could be predicted from null-cline analysis. This allowed us to find the bistable and tristable regions of the parameter space, the regions where no germination is expected in the deterministic limit and the regions where

**Table 2.** Default values for each parameter in our mathematical model and varying parameters used in figures showing 1D parameter scans.

$\beta_X$: production rate for $X$; $\nu_X$: degradation rate for $X$; $\theta_{X,Y}$: threshold above which $Y$ has an effect on $X$; $C_{X,Y}$: coefficient for regulatory functions of $Y$ acting on $X$; $h$: exponent of regulatory functions; $V$: effective system volume (modulates noise). All parameter units are arbitrary.

| | Default | Figure 5B–D | Figure 5—figure supplement 2A | Figure 5—figure supplement 2B | Figure 5—figure supplement 2C | Figure 5—figure supplement 2D |
|---|---|---|---|---|---|---|
| $\beta_{ABA}$ | 1 | | Varying | | | |
| $\beta_{GA}$ | 0.3 | | | | | |
| $\beta_{GA,Z}$ | 0.01 | | | | | |
| $\beta_Z$ | 39 | | | | | |
| $\beta_I$ | 0.3 | | | | | |
| $\nu_{ABA}$ | 1 | | 1.58 | | | |
| $\nu_{GA}$ | 1 | | | | | |
| $\nu_Z$ | 0.1 | | | | | |
| $\nu_I$ | 0.4 | | | | | |
| $\theta_{ABA,I}$ | 3.7 | | | | | |
| $\theta_{GA,I}$ | 1.2 | | | | | |
| $\theta_{I,ABA}$ | 6.5 | Varying | | | 10 | 10 |
| $\theta_{I,GA}$ | 6 | | | Varying | Varying | |
| $C_{ABA,I}$ | 10 | | | | | |
| $C_{GA,I}$ | 4 | | | | | |
| $C_{I,ABA}$ | 10 | | | | | |
| $C_{I,GA}$ | 6 | | | | | |
| $h$ | 4 | | | | | |
| $V$ | 30 | | | 100 | | Varying |

germination would instantaneously occur. The remaining regions in the parameter space were monostable. Note that monostable, bistable and tristable regions were computed by counting the number of steady states after the rise of GA production. Regions where no germination would occur in the deterministic limit were those where the lowest fixed point for the integrator was higher than the germination threshold. Regions where germination would instantaneously occur were those regions having the highest Integrator fixed point below the germination threshold before the sowing-induced increase of GA production. CV and mode of the simulations were represented when there were more than nine seeds germinating out of 1000, so the percentage of germination was equal to or higher than 1%.

The theoretical regions across the parameter spaces, obtained from nullcline analysis of the deterministic model, closely predicted the stochastic simulation outcomes (*Figure 5—figure supplement 4*). One exception was that simulations occasionally showed some germination happening after a small number of simulation steps in the instantaneous germination region. In these cases, even if initial conditions were below the germination threshold, after an integrator step, the Integrator variable was not below the threshold anymore due to the stochastic fluctuations, and therefore, germination happened later on during the simulation (*Figure 5—figure supplement 4F*). Also, in the area where no germination was expected in the deterministic limit, germination could occur in some occasions due to stochastic fluctuations, leading to different germination percentages. This happened either when the germination threshold was just below the lowest Integrator fixed point after the rise of GA production or at high noise intensities (e.g. see *Figure 5—figure supplement 4E,G*).

Our theoretical analysis and simulations showed that there are two different prototypical dynamical behaviours that are most biologically relevant (*Figure 5—figure supplement 1*). On one side, simulations in which there is bistability after the rise of GA production, where seeds can undergo a transient in which they remain in a high Integrator state above the germination threshold, until stochastic fluctuations make them switch to the low Integrator state, driving germination. On the other side, simulations in which there is monostability after the rise of GA production, where the seeds achieve the low Integrator state in a more direct manner. Those simulations falling within the instantaneous germination region would not be biologically relevant, given that a certain time is needed for seeds to germinate after sowing. Simulations leading to germination and falling within the non-germination region in the deterministic limit would also be less biologically relevant, given that the Integrator would repeatedly cross the threshold back and forth, not persisting below it.

Note the instantaneous germination regions and the non-germination regions can also contain monostable, bistable and tristable cases. For simplicity, throughout the text, we use the term monostable regions to refer to those regions in the parameter space that are monostable after the rise of GA production and do not overlap with these two less biologically relevant regions (i.e. such biologically relevant monostable regions correspond to the white theoretical predicted regions in *Figure 5—figure supplement 4*).

The parameter space explorations and derived panels in *Figure 5—figure supplements 2–5* show stochastic simulation runs for 400 seeds. The exploration of the ABA sensitivity parameter (*Figure 5*) and the dose dependence plots and derived panels shows simulation runs for 4000 seeds (*Figure 6*, *Figure 6—figure supplements 1* and *2*), as do simulations shown in *Figure 5—figure supplement 1*.

In most of the presented simulated results we have excluded those simulation points corresponding to parameter values that could lead to germination even without the rise of GA production (i.e. for $\beta_{GA,Z} = 0$). These excluded parameters would represent the possibility of having germination prior to sowing, which is not biologically relevant. In particular, additional stochastic simulations on 4000 seeds were run in relation to *Figure 5* to exclude such non-biologically relevant points, and simulations on 40 seeds were run in relation to *Figure 5—figure supplements 2* and *5* for the same purpose. Additionally, we corroborated that no germination occurred if we eliminated the rise of GA production for the simulation cases studied in *Figure 5—figure supplement 1*, and for the cases in which no exogenous ABA and GA was applied in *Figure 6* and *Figure 6—figure supplement 1*. To do so, in each of these cases, a stochastic simulation on 4000 seeds was run with $\beta_{GA,Z} = 0$. In all these figures, we also excluded those points falling within the less biologically relevant theoretical regions mentioned above, that is, the non-germination region in the deterministic limit, and the spontaneous germination region – and we found that parameter values falling in the spontaneous

germination region were a subset of parameters where there would be germination even without the rise of GA production. When applying exogenous ABA or GA in *Figure 6* and *Figure 6—figure supplement 1*, we performed appropriate checks, checking that both controls and the applied doses were not fulfilling the spontaneous germination condition (as indicated by the level of the highest unperturbed fixed point of the Integrator) and that the controls did not germinate without the rise of GA production, as explained above. We also corroborated that both the controls and the applied doses would not fall in the non-germination condition in the deterministic limit. In the 2D parameter explorations (*Figure 5—figure supplement 4*), we represented all the simulated points, such that all behaviours (biologically relevant and less biologically relevant) could be studied and exemplified.

Numerical simulations were performed with the Organism simulator (https://gitlab.com/slcu/teamHJ/Organism; *Jonsson et al., 2005*). Modelling figures were produced with the Matplotlib Python library (*Hunter, 2007*).

## Data availability statement

Whole-genome sequence data was deposited to NCBI's Short Read Archive (BioProject accession PRJNA486286). All data analysis and modelling scripts can be found at https://gitlab.com/slcu/teamJL/abley_formosa_etal_2020. Both the raw and processed experimental data for use with the analysis scripts are available from the Cambridge Apollo Repository (https://doi.org/10.17863/CAM.66984).

## Acknowledgements

We thank Sandra Cortijo for critical reading of the manuscript and Casandra Villava, Ting Wang and Helena Kelly for help with taking care of plants and seed scoring. P F-J. thanks Ruben Perez-Carrasco for fruitful discussions about the modelling. Work in the Locke and Leyser labs was supported by fellowships from the Gatsby Charitable Foundation (Locke lab: GAT3272/GLC and Leyser Lab: GAT3272C).

## Additional information

### Funding

| Funder | Grant reference number | Author |
| --- | --- | --- |
| Gatsby Charitable Foundation | GAT3272C | Ottoline Leyser |
| Gatsby Charitable Foundation | GAT3272/GLC | James CW Locke |

The funders had no role in study design, data collection and interpretation, or the decision to submit the work for publication.

### Author contributions

Katie Abley, Conceptualization, Data curation, Formal analysis, Supervision, Validation, Investigation, Methodology, Writing - original draft, Writing - review and editing, carried out experiments; Pau Formosa-Jordan, Conceptualization, Formal analysis, Supervision, Investigation, Visualization, Methodology, Writing - original draft, Writing - review and editing, carried out modelling; Hugo Tavares, Data curation, Formal analysis, Investigation, Methodology, Writing - original draft, Writing - review and editing, carried out QTL mapping and sequence data analysis; Emily YT Chan, Formal analysis, Investigation, carried out modelling; Mana Afsharinafar, Investigation, carried out experiments; Ottoline Leyser, Conceptualization, Supervision, Funding acquisition, Writing - review and editing; James CW Locke, Conceptualization, Supervision, Funding acquisition, Writing - original draft, Writing - review and editing.

### Author ORCIDs

Katie Abley (iD) https://orcid.org/0000-0001-5524-6786

Pau Formosa-Jordan (iD) https://orcid.org/0000-0003-3005-597X

Hugo Tavares [iD] http://orcid.org/0000-0001-9373-2726
Ottoline Leyser [iD] https://orcid.org/0000-0003-2161-3829
James CW Locke [iD] https://orcid.org/0000-0003-0670-1943

Decision letter and Author response
Decision letter https://doi.org/10.7554/eLife.59485.sa1
Author response https://doi.org/10.7554/eLife.59485.sa2

## Additional files

### Supplementary files

- Supplementary file 1. Supplementary_File1_MAGICs.
- Supplementary file 2. Supplementary_File2_MAGICParents.
- Supplementary file 3. Supplementary_File3_SpanishAccessions.
- Supplementary file 4. Genotyping primers.
- Transparent reporting form

### Data availability

Whole genome sequence data was deposited to NCBI's Short Read Archive (BioProject accession PRJNA486286). All data analysis and modelling scripts can be found at https://gitlab.com/slcu/teamJL/abley_formosa_etal_2020, copy archived at https://archive.softwareheritage.org/swh:1:rev:0a97b841e58b128c174d93fc759b28f1df2966a2. Both the raw and processed experimental data for use with the analysis scripts are available from the Cambridge Apollo Repository (https://doi.org/10.17863/CAM.66984).

The following datasets were generated:

| Author(s) | Year | Dataset title | Dataset URL | Database and Identifier |
|---|---|---|---|---|
| Abley K, Tavares H, Formosa-Jordan P, Chan EY, Afsharinafar M, Leyser O, Locke JCW | 2021 | Data supporting 'An ABA-GA bistable switch can account for natural variation in the variability of Arabidopsis seed germination time', Abley & Formosa-Jordan et al., eLife (2021) | https://doi.org/10.17863/CAM.66984 | Cambridge Apollo Repository, 10.17863/CAM.66984 |
| Abley K, Formosa-Jordan P, Tavares H, Chan E, Leyser O, Locke JCW | 2018 | Genetic basis for variability in Arabidopsis seed germination time | https://www.ncbi.nlm.nih.gov/bioproject/?term=PRJNA486286 | NCBI BioProject, PRJNA486286 |

The following previously published dataset was used:

| Author(s) | Year | Dataset title | Dataset URL | Database and Identifier |
|---|---|---|---|---|
| Kover PX, Valdar W, Trakalo J, Scarcelli N, Ehrenreich IM, Purugganan MD, Durrant C, Mott R | 2009 | A Multiparent Advanced Generation Inter-Cross to Fine-Map Quantitative Traits in Arabidopsis thaliana | http://mtweb.cs.ucl.ac.uk/mus/www/magic/ | NA, NA |

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
