## [Decision Letter]

**Acceptance summary:**

This work begins to study how non-linear regulatory systems and natural variation intersect to control key fitness traits. They show that natural variation in a bistable switch relationship between ABA and GA, two key plant hormones, influences the natural variation and stochasticity in seed germination timing within *Arabidopsis thaliana*.

**Decision letter after peer review:**

Thank you for submitting your article "An ABA-GA bistable switch can account for natural variation in the variability of Arabidopsis seed germination time" for consideration by *eLife*. Your article has been reviewed by three peer reviewers, one of whom is a member of our Board of Reviewing Editors, and the evaluation has been overseen by Christian Hardtke as the Senior Editor. The reviewers have opted to remain anonymous.

The reviewers have discussed the reviews with one another, and the Reviewing Editor has drafted this decision to help you prepare a revised submission.

Summary:

In this work, the authors proceed to begin investigating the basis of stochastic variation in seed germination using modelling and natural variation.

Essential revisions:

All the reviewers and editors agreed that the work presented was very interesting. However, the work feels like two separate manuscripts, one on modelling and one on natural variation. The editors and reviewers judged that there needs to be some form of more explicit connection between the two parts. Each reviewer provided some ideas on how to accomplish this including some that revolve around solely analyzing the existing data.

There is also a desire to be more introspective about the discussion of coupling between mean and CV of this trait as there is a genetic and mechanistic connection. For example, there is a correlation between mean and CV but this by no means explains all the data. As such, they are not independent, but neither are they fully dependent. An acknowledgement of this throughout and possibly finishing with saying that more work is needed to find and test how many loci are unique to mode or CV versus shared between the two would be sufficient.

Finally, we are not fully convinced by the bet-hedging experiment. It does not seem to have much connection to biology and is more a proof that if plants are alive, they are selected against. This experiment is not really essential to the manuscript and could be dropped. Or possibly perform an ecological analysis of haplotype diversity across Europe to provide more biological relevance to this claim.

*Reviewer #1:*

This is a very interesting manuscript looking at CV in germination timing from a number of angles including natural variation and modelling. There are some terminology and statistical issues that complicate the manuscript, but these are readily addressable. The biggest struggle I had was that there is a disconnect between the MAGIC and Modelling work. Essentially the MAGIC ends with saying it might be one ABA and one GA related gene controlling the variation in CV and the Modelling instantly jumps into the ABA/GA. While this is interesting, there is not a strong link in the end between the two sections. In Figure 7, the authors use the ABA/GA concentrations on the high and low variability lines but then never really go back and use this data to show that these lines are altering the expected components in the model. This could be readily tested by looking at the haplotypes at the two QTL in these lines, incorporating them into the ABA/GA model and comparing to the results. For example, do the haplotypes predict the unique response to GA in Figure 7B-iv? Instead, the authors simply treat them as high vs low lines rather than diving down to the haplotypes at the two QTLs. Without this connection, the manuscript feels like a set of disconnected vignettes. While I provide one instance of how to link the model to haplotype variation there are other possibilities, and this should be attempted.

The authors discuss in the Introduction that bet-hedging is solely where the same genotype produces different phenotypes, but this is simply one form of bet-hedging. It is also possible to achieve bet-hedging by the presence of standing genetic variation within a population as a way to respond to fluctuating or other non-linear selective processes. It would help to clarify that the authors are focused on stochastic or probabilistic bet-hedging and not the other forms.

The authors focus solely on how bet-hedging may enable the avoidance of stress conditions. But there is another possibly equally important effect of lengthening the generation time which allows the long-term maintenance of genetic variation required for stressful conditions. There is literature on how longer generation times provide a form of "memory" to maximize the ability to maintain the potential to have genetic variation required for unique selective events. In this model, delayed germination would lengthen the generation time and actually work to maintain germplasm that is adapted to rare stressful conditions.

Discussing the variation of variation is not the easiest thing to write and there are places where the text is not very clear. For example, the sentence "Additionally, the extent to which there is natural variation in germination time distributions in Arabidopsis has not been described." Is meant to be specific about mapping the distribution as the trait but it is kind of fuzzy and the authors of the DOG studies may take umbrage with sentences like this. I would suggest developing a single terminology for variance as a trait and use it consistently. I understand that it is not the most pleasant in a grammatical context to repeat terminology but given that the community is not used to variance as a trait this will greatly help the audience to really understand what is being communicated. For example, "variability in germination time" could equally be referring to natural genetic variation that alters the mean or the stochastic component of the trait.

The seeds from 3 plants were collected. Were they phenotyped separately or were the 3 plants mixed?

I am a bit puzzled by this sentence "We chose a relatively short period of dry storage for our experiments as we reasoned that this would best reflect the state of naturally shed seeds". Is there evidence that most Arabidopsis accessions rapidly re-germinate within a month? This would only be true in some specific environments, correct? For example, seeds in Mediterranean environments should have a long delay to ensure that they germinate at the start of the next rains. And Scandinavian summer accessions would equally over-winter as seeds. Even going to 60 days in these environments does not capture the spectrum. I agree that this is a choice but am not sure that it is valid to say it reflects the "natural state". The authors had to choose one environment. I would simply state this and that this is the one chosen, and it would be useful to try other environments in the future.

In the Results, it appears that the authors have conducted independent biological experiments of maternal growth and seed set. This provides the ability to directly quantify heritability of CV rather than the more visual analysis that has been done. The authors should utilize a linear model to directly estimate the heritability of CV in this trait to compare with other traits where CV heritability has been measured.

In Figure 1—figure supplement 1 part A/B, the authors should not include a line of 1:1 unless they have explicitly tested for a relationship. Given this analysis, the authors should conduct a mixed linear ANCOVA with genotype and seed age to directly quantify the influence of the two factors on the CV. Given this analysis, it is likely that they will find that the slope of CV is decreasing across storage time and as such, storage is having an influence on the CV. This will also allow them to show that the rank order of genotypes is not greatly changing across time.

In Figure 2B, it is not clear what these represent. Are these all the siliques on a single plant arranged from top to bottom or are they single siliques from the same plants shown in part A? Similarly, I'm not sure how the top and bottom half of a single silique provides sufficient information to claim that there is no variation along an inflorescence. And by bottom/top, is this distance from the maternal rosette or from the inflorescence stem? The gradients in a single silique would be largely driven by the distance from the inflorescence stem.

As a general statistical guideline, whenever a regression line is shown in a plot, the figure legend should have the information surrounding the line, method, R2, N and significance. Additionally, most of these regressions are conducted using Pearson which is very sensitive to abnormal distributions which are present in most of these data. These should most likely be conducted using Spearman Rank or other similar non-parametric tests.

I'm unsure why the authors correlated the parental effects to the haplotype effects for the two QTLs in a system where there is transgressive segregation and likely epistasis. The use of this correlation is only valid if the model is totally additive with no transgression. In this system, this data doesn't provide evidence one way or another. At best they could develop a two-locus model using the haplotypes and run them as a model together to look at additivity and epistasis to try and explain the parents. But running as single loci isn't informative.

The bet-hedging experiment seems unconnected to the natural variation. Essentially this treatment is a 100% selection event that is never going to occur in nature. The experiment is set up to test if non-germinated seedlings provide the ability to evade absolute death that occurs at a specific date prior to the latest germination time. As such, this is less an experiment and more a control that the phenotyping is reproducible. I'm not sure if this really adds to the manuscript as is. It would need to incorporate some aspect of the natural variation like are the high CV parents more likely to be in fluctuating environments, etc to help support the argument.

Reviewer #2:

Abley et al., use seed germination from the model plant Arabidoss to study, at the genetic level, phenotypic variability. For this, authors used 19 natural accessions and a multi-parent recombinant inbred population. QTL analysis led to the identification of two loci, one in chromosome 3 and the other in chromosome 5. Genes related to ABA signaling such as DOG1 and DOG6 are part of the loci. Mathematical modelling was needed to explain how such genetic loci could influence germination time. A stochastic model involving a bistable switch of players from ABA and GA pathways, led authors to test the effects of disturbing, genetically or by exogenous application, ABA and GA levels, on germination. Finally, authors suggest that variation in seed germination is a strategy of the plant survive upon drastic abiotic stress.

If we consider the study in several consecutive parts such as A. The analysis on germination times of the diverse genetic backgrounds which led to B. a QTL analysis to find genes that were already known to influence germination, such as SOG1 and SOG6, that led to C. Modelling involving, very few by the way, players in ABAignalling pathways to explain D. that GA and ABA levels are key to understand phenotypic variation on germination time.

1) My main concern is how can authors explain the need of the model in this story? since based on their data in B they could have gone directly to D.

2) Moreover, the stochastic model suggested to test the levels of GA and ABA, but even if they sustain in their response the need of the model, how can they connect the data in B with the need to explore, involve, GA in the equation,?

3) Authors should elaborate how they justify the inclusion of GA signalling in the model, if QTL data did not suggest this. Of course, they can claim that GA and ABA are well known players in seed germination, but still the way it is written it feels that GA appears suddenly in the story.

4) If the tracked phenotype is related to loci that involves SOG1, SOG6 and SET1

genes, why authors did not generate, or use mutants in these genes to explore seed germination time? this is one of my major questions, and the only extra experiment work that this reviewer would like to see in a future version of the manuscript.

Reviewer #3:

In this manuscript, Abley and colleagues map differences in germination phenotypes, including the coefficient of variation (CV), the mode, and the percent germinated, in the *A. thaliana* MAGIC line population, finding two QTL. They then develop a computational model for germination incorporating the integration of ABA and GA pathways to generate a bistable switch for the germination decision and demonstrate that the model can output a variety of CV, mode, and germination percentage phenotypes. Using the model, they then predict and test the response of MAGIC lines with high and low CV to exogenous ABA and GA applications.

I agree with the general premise that bet-hedging and CV variation are potentially ecologically and evolutionarily important and I appreciate the development of a model that serves as a basis for understanding both the molecular and natural genetic mechanisms that generate germination phenotypes. My major concern with this manuscript, however, is that it doesn't quite meet its promise in showing that the bistable switch proposed by the model underlies natural variation in the phenotypes measured here. What is demonstrated is that the model can (possibly) account for the effect of endogenous ABA or GA on MAGIC lines with different CVs. Strictly speaking this isn't the same as demonstrating that the model explains how this CV variation in the MAGIC (or parental lines) is generated to begin with. To do this, one would have to have specific information/predictions about how underlying genetic variation affected model components and then demonstrate that THESE modulations could generate the CV variation observed in parental lines (or MAGIC lines). For example, let's say that a particular allele at the QTL on chromosome three caused reduced expression of DELLAs, subsequently changing the integrator parameter of the model. Would incorporating this modified parameter explain the CV variation between plants with the different variants of this QTL? In some cases, the manuscript makes the distinction between the response and origin questions clear, but in key areas (like the title and abstract), this difference gets blurred.

1) The experiment meant to demonstrate that there is no pattern of germination variability along the plant is not very conclusive. Showing single silique distribution patterns with siliques ordered basipetally would more convincingly demonstrate that there is no relationship between silique age and CV.

2) I am not convinced that mode, CV, and percent germination can be genetically uncoupled. For the most part, there are fairly strong relationships between these phenotypes in the MAGIC lines, with a few (maybe 10 or so) outlying lines where the CV is (presumably) much higher than would be predicted from the mode or percentage germination (Figure 4A – by the way, what is the overlap between these outliers and with the lines with bimodal CV distributions?). The data demonstrating uncoupling consists of several cherry-picked examples in Figure 3 (and its associated supplements). One would really need to look at this more systematically to demonstrate the extent to which this decoupling is actually significant (i.e. considering the variability in mode and CV estimates between replicates, are there lines in which the relationship between CV and mode is different from the predicted relationship?). And in fact, the QTL results also suggest that these phenotypes can't be uncoupled – any QTL for mode or percent is also a QTL for CV, and the QTL on chromosome 5 that the authors claim is CV-specific both explains little variation and its marginal significance depends on whether or not 8 particular lines with bimodal distributions (and therefore perhaps cases in which mode doesn't describe centrality well) are included in the analysis. Even in the bulk segregant analysis, there is a clear relationship between germination time of F2s parents and CV of subsequent F3s. I therefore think that the strength of this claim needs to be reconsidered.

3) Comparing Figure 6 and Figure 7 (the predicted and observed responses of GA and ABA in high and low CV lines), there seem to be quite a number of cases in which the predicted response is different from that observed (the mean and mode in response to GA for example) or in which there is extremely high variability among the lines tested, with only a subset of lines matching the predicted responses (the CV and mean response of high CV lines to ABA, for example). So, at what point can one say that the predicted response is consistent with the observed one? And why would one expect such high variability between lines? Do they harbor QTL alleles that would explain these differences? This might be one way to tie the mapping and modeling parts of the manuscript together better (they currently kind of feel like two disconnected stories – one doesn't need to have mapped QTL to generate the model or confirm the response of high/low CV lines to GA and ABA).

4) The experiment meant to demonstrate the adaptive value of bet hedging is quite trivial – one doesn't really need an experiment to show that killing more plants in one genotype will result in fewer plants of that genotype – and therefore I think the importance of result is overstated in the manuscript. Demonstrating that differences in CV have adaptive value really would be a difficult experiment to do and would need to incorporate some sort of natural (or very closely simulated natural) conditions. If such experiments are attempted, I would also strongly recommend NOT using Col as one of the accessions. Col is presumably adapted to lab conditions, not natural ones, and it is easy to think of how early and low CV germination would be selected for in a laboratory setting (especially if CV can be impacted by a 1-2 major loci).

---

## [Author Response]

Essential revisions:All the reviewers and editors agreed that the work presented was very interesting. However, the work feels like two separate manuscripts, one on modelling and one on natural variation. The editors and reviewers judged that there needs to be some form of more explicit connection between the two parts. Each reviewer provided some ideas on how to accomplish this including some that revolve around solely analyzing the existing data.

We thank the reviewers for their suggestions on how we could form a better link between the natural variation and modelling parts. We have explored all of the suggestions given and have made changes to more explicitly link the genetics and natural variation with the modelling. In the revised version presented here, we explore the hypothesis that the difference between high and low variability lines lies in differences in their ABA sensitivity. This hypothesis is suggested by the candidate genes underlying the QTL peaks, which we have tested in this revised version, providing evidence for their roles in ABA sensitivity during germination and for effects on germination time distributions (see response to reviewer 2, comment 2). We show that, in the model, varying ABA sensitivity can capture differences in variability and the positive correlation in the MAGIC lines between CV and mode.

We simulate high and low variability MAGIC lines, and the ABA and GA dose responses applied to them, with a difference in ABA sensitivity and provide experimental evidence for higher ABA sensitivity in the high variability MAGIC lines.

We also show that other parameters in addition to ABA sensitivity could account for correlated variation in CV and mode, and that some parameters can account for uncoupled variation in CV and mode (and so genetic variation for these parameters could explain the weak coupling between CV and mode seen in the MAGIC lines).

We provide more insight into the basis of high variability in the model, and show that variation in ABA sensitivity, and in some other parameters, affects CV by influencing whether the model operates in a monostable or bistable regime and by influencing the stability of the non-germination steady state within the bistable regime. In our modelling approach, we assume that all seeds of all lines exist in a stable non-germinating state prior to sowing and that sowing (i.e. exposure to water and light) causes a rise in the rate of GA biosynthesis, enabling this situation to change. After this rise in GA biosynthesis, we explore cases where the model operates either in a monostable regime where germination is the only possible state (i.e. the non-germinating state is lost), or a bistable regime, where seeds may either germinate or not. With low ABA sensitivity the model is monostable after sowing, with the high GA, low ABA, low integrator (germination) steady state the only stable state after sowing and the rise in GA biosynthesis. This means that all seeds switch rapidly from their initial non-germinating state (as it disappears upon the rise in GA biosynthesis) into the germination state. With higher ABA sensitivity, the model is bistable after sowing and the rise in GA biosynthesis, with a low GA, high ABA, high integrator (non-germination) steady state in addition to the germination steady state both existing after sowing. After the rise in GA biosynthesis, seeds can remain in the non-germination steady state for some time but are driven to transition to the germination steady state by stochastic fluctuations, which results in variable germination times. When operating within the bistable regime, the stability of the nongermination steady state influences the level of variability such that when this state is more stable, if germination still occurs, it becomes more variable. Hence, our overall hypothesis is that the natural variation between MAGIC lines in the variability of germination time is caused by (i) the ABA-GA network operating in different regimes (i.e. monostable vs bistable) in different lines, and (ii) differences between lines in the stability of the non-germination steady state when bistability occurs following sowing.

There is also a desire to be more introspective about the discussion of coupling between mean and CV of this trait as there is a genetic and mechanistic connection. For example, there is a correlation between mean and CV but this by no means explains all the data. As such, they are not independent, but neither are they fully dependent. An acknowledgement of this throughout and possibly finishing with saying that more work is needed to find and test how many loci are unique to mode or CV versus shared between the two would be sufficient.

We have updated the text throughout to de-emphasise the extent to which the traits are decoupled. We agree with the reviewer that it is likely that there is a genetic and mechanistic connection between CV and mode/mean and we now argue that this coupling can be accounted for by assuming that the major difference between high and low variability lines is related to ABA sensitivity.

We have changed the subsection to: “Variability is weakly coupled to modal germination time and percentage germination”.

In the Results we conclude: “Thus, within the MAGIC population, variability is correlated with percentage germination and modal germination time but can be uncoupled from these traits.”

We have changed the titles of Figure 3 and Figure 3—figure supplement 1 to read: “Variability is weakly coupled to x trait.”

We have changed the subsection to read: “QTL mapping in MAGIC lines reveals two QTL underlying variability in germination time”, thus de-emphasising the claim that the Chr5 QTL controls CV independent from other traits. We’ve also made changes in the text of this Results section so that we no longer claim the Chr5 QTL is variability specific, just that we couldn’t detect statistically significant effects for mode, mean and percent germination. We conclude this section with: “In summary, we have shown that at least two loci contribute to variability in seed germination time in the MAGIC lines (chromosome 3, ~16 Mb and chromosome 5, ~18.6/ 19.8 Mb). Consistent with a correlation between CV, mode days to germination and percentage germination in the MAGIC lines, the main QTL on chromosome 3 has correlated effects on all these three traits. The locus at ~19 Mb on chromosome 5 appears to affect variability most strongly.”

We also now discuss that given that the chromosome 5 QTL is most likely caused by the *DOG1* gene (or possibly *AHG1*), which are both general regulators of germination, the lack of a detectable effect on mode could be because the alleles of these genes present in the MAGIC lines most strongly effect CV or could be due to us not having enough statistical power to detect an effect on this trait. We explore this in the Discussion, where we state:

“In the QTL mapping, the chromosome 5 locus was only found to be significantly associated with the CV of germination time. However, besides their described effect on CV, the mutant alleles of the two candidate genes we tested, *DOG1* and *AHG1*, also affected mode and percentage germination (Née et al. 2017; Bentsink et al. 2006). The fact that we did not detect an association between this region of chromosome 5 and those two latter traits could be explained by differences between the naturally occurring alleles within the MAGIC population (which may predominantly affect CV) and the mutant alleles of these genes (which affect multiple traits). It could also be the case that the MAGIC lines’ alleles affect multiple germination traits but that we didn’t have enough statistical power to detect this.”

Finally, we are not fully convinced by the bet-hedging experiment. It does not seem to have much connection to biology and is more a proof that if plants are alive, they are selected against. This experiment is not really essential to the manuscript and could be dropped. Or possibly perform an ecological analysis of haplotype diversity across Europe to provide more biological relevance to this claim.

We have removed the bet-hedging experiment.

Reviewer #1:This is a very interesting manuscript looking at CV in germination timing from a number of angles including natural variation and modelling. There are some terminology and statistical issues that complicate the manuscript, but these are readily addressable. The biggest struggle I had was that there is a disconnect between the MAGIC and Modelling work. Essentially the MAGIC ends with saying it might be one ABA and one GA related gene controlling the variation in CV and the Modelling instantly jumps into the ABA/GA. While this is interesting, there is not a strong link in the end between the two sections. In Figure 7, the authors use the ABA/GA concentrations on the high and low variability lines but then never really go back and use this data to show that these lines are altering the expected components in the model. This could be readily tested by looking at the haplotypes at the two QTL in these lines, incorporating them into the ABA/GA model and comparing to the results. For example, do the haplotypes predict the unique response to GA in Figure 7B-iv? Instead, the authors simply treat them as high vs low lines rather than diving down to the haplotypes at the two QTLs. Without this connection, the manuscript feels like a set of disconnected vignettes. While I provide one instance of how to link the model to haplotype variation there are other possibilities, and this should be attempted.

We thank the reviewer for their positive assessment of our work, as well as their very useful suggestions for making stronger connections between the experimental and modelling sections of the manuscript. Please see response to editor’s comment 1 where we describe how we have made a more explicit hypothesis for the mechanism underlying the differences between high and low variability lines.

In relation to the reviewer’s suggestion about looking into the haplotypes at the two QTL for the lines used for the ABA and GA dose responses, we have included Table 1, which shows which haplotypes the high and low variability MAGIC lines have at the two QTL and the estimated effects of these haplotypes on CV. This shows that the high and low variability MAGIC lines we chose for the experiment separate most clearly by the Chr3 haplotypes: all of the high variability MAGIC lines have Chr 3 haplotypes associated with medium or high CV and all of the low variability MAGIC lines have Chr3 haplotypes associated with medium or low CV (and although there are haplotypes of “medium” effect in both high and low variability groups, all the haplotypes in the high variability lines were associated with higher effect on CV than those in the low variability lines) (see Figure 4—figure supplement 2A for haplotype effects). The overall tendency is for all of the low variability lines to behave similarly to each other, and the same for the high variability lines, despite the presence of different haplotypes at the 2 QTL amongst each group of lines. We would need to perform the ABA and GA dose responses for a much larger number of lines systematically chosen for different haplotype combinations to make inferences about how individual haplotypes are related to the response to ABA or GA. Or alternatively, we could analyse haplotype effects in transgenic lines where mutants in Col-0 were complemented with different accession haplotypes (see last comment of this section). We feel that these experiments are beyond the scope of this paper. But the ABA dose response experiment presented here shows that the high variability lines are more sensitive to ABA than the low variability lines (Figure 6B iv), supporting our hypothesis that the difference between high and low variability lines lies in their ABA sensitivity. Since we now think that the best parameter to represent the effects of candidate genes for both QTL is that controlling the sensitivity of integrator production to ABA, we have changed how we model the high and low variability lines and represent them with high and low sensitivities to ABA, respectively.

The behaviour of line 285 cannot be explained by its haplotype effects at these QTL. This line has chromosome 3 and chromosome 5 haplotypes associated with high variability, as does line 393, but the responses of the two lines are different. As we are not able to account for the MAGIC lines with unique responses to ABA/GA with their Chr3 and Ch5 haplotypes (i.e. the haplotypes were not informative in explaining the response) we didn’t attempt to model each MAGIC line with a unique parameter value.

In the discussion we state that in the future it will be important to investigate further the role of specific haplotypes, stating:

“Further work will be needed to identify the causal variants underlying both of the QTL identified here and to determine the extent to which they have correlated or uncoupled effects on different germination traits. It will be important to test directly candidate genes through transforming candidate gene null mutants with corresponding alleles cloned from high and low variability lines (Weigel, 2012). These transgenic lines could then be used to understand how natural variation at the two QTL influences germination time distributions, by using them to quantify the effects of the different accession alleles on CV, mode and percentage germination as well as on traits related to the ABA-GA network such as ABA and GA sensitivity.”

The authors discuss in the Introduction that bet-hedging is solely where the same genotype produces different phenotypes, but this is simply one form of bet-hedging. It is also possible to achieve bet-hedging by the presence of standing genetic variation within a population as a way to respond to fluctuating or other non-linear selective processes. It would help to clarify that the authors are focused on stochastic or probabilistic bet-hedging and not the other forms.

We agree with the reviewer that there is more than one way for a plant population to hedge its bets, and so now clarify the type of bet-hedging we are interested in in the introduction as the reviewer suggests. We have introduced the possibility of phenotypic diversity being produced through maintenance of genetic diversity and clarified that we are focused on differences between genetically identical individuals. Our Introduction now starts as follows:

“In an environment where current cues cannot be used to predict future conditions, exhibiting a range of phenotypes may promote species survival. A population can hedge its bets against an uncertain environment by maintaining genetic variation within the population (e.g. balancing selection (Delph and Kelly, 2014)) or by containing a single genotype that produces a variety of phenotypes (a diversified bethedging strategy (Cohen, 1966, 1966; Lewontin and Cohen, 1969, 1969; Philippi and Seger, 1989; Simons, 2011, 2011)). Bacteria can use diversified bet-hedging strategies to survive a range of changes of condition, including antibiotic treatments (Balaban et al., 2004; Martins and Locke, 2015) and other environmental stresses (Patange et al., 2018). Mechanistic studies suggest that the required phenotypic variability between genetically identical individuals is generated by genetic networks amplifying stochasticity in molecular interactions to generate a range of outputs (Alon, 2007; Eldar and Elowitz, 2010; Viney and Reece, 2013).

In plants, theoretical work shows that variability in the seed germination time of genetically identical seeds is likely to be advantageous in environments that are unpredictable (Cohen, 1966; Simons, 2011).”

And later in the Introduction, we state: “Here we set out to investigate the causes of germination time variability among genetically identical Arabidopsis seeds”.

The authors focus solely on how bet-hedging may enable the avoidance of stress conditions. But there is another possibly equally important effect of lengthening the generation time which allows the long-term maintenance of genetic variation required for stressful conditions. There is literature on how longer generation times provide a form of "memory" to maximize the ability to maintain the potential to have genetic variation required for unique selective events. In this model, delayed germination would lengthen the generation time and actually work to maintain germplasm that is adapted to rare stressful conditions.

We agree that other functions are possible for variable germination times, and that variability in germination time, like seed dormancy, could contribute to the long-term maintenance of genetic diversity by contributing to seed bank formation. We have added the following to the Discussion:

“Alternative functions beyond bet-hedging can be explored, including the possible function of distributed germination times in avoiding seedling competition, or that the variable germination could contribute to the seed bank, which acts as a persistent source of genetic diversity that may be advantageous for longterm species survival in variable environments (Templeton and Levin 1979).”

Discussing the variation of variation is not the easiest thing to write and there are places where the text is not very clear. For example, the sentence "Additionally, the extent to which there is natural variation in germination time distributions in Arabidopsis has not been described." Is meant to be specific about mapping the distribution as the trait but it is kind of fuzzy and the authors of the DOG studies may take umbrage with sentences like this. I would suggest developing a single terminology for variance as a trait and use it consistently. I understand that it is not the most pleasant in a grammatical context to repeat terminology but given that the community is not used to variance as a trait this will greatly help the audience to really understand what is being communicated. For example, "variability in germination time" could equally be referring to natural genetic variation that alters the mean or the stochastic component of the trait.

We agree with the reviewer that the terminology involved can be confusing, and we thank them for their suggestions. In the Introduction, we have clarified our use of terms:

“Hereafter we use the term “variability” to refer to phenotypic differences between genetically identical individuals, and “variation” to refer to differences between genetically distinct individuals, such as natural variation between accessions.”

We have changed the sentence that the reviewer points out as ambiguous and now state: “it is unclear whether different Arabidopsis genotypes generate different degrees of variability in germination times”.

The seeds from 3 plants were collected. Were they phenotyped separately or were the 3 plants mixed?

We have made this more explicit in the Materials and methods:

“Following filtering, for 91% of MAGIC lines, at least three replicate seed batches (each collected from a different parent plant) were used for each genotype. The seed batches were sown separately, with one petri dish for each of the three batches.”

I am a bit puzzled by this sentence "We chose a relatively short period of dry storage for our experiments as we reasoned that this would best reflect the state of naturally shed seeds". Is there evidence that most Arabidopsis accessions rapidly re-germinate within a month? This would only be true in some specific environments, correct? For example, seeds in Mediterranean environments should have a long delay to ensure that they germinate at the start of the next rains. And Scandinavian summer accessions would equally over-winter as seeds. Even going to 60 days in these environments does not capture the spectrum. I agree that this is a choice but am not sure that it is valid to say it reflects the "natural state". The authors had to choose one environment. I would simply state this and that this is the one chosen, and it would be useful to try other environments in the future.

We have removed the sentence that the reviewer highlights. We have added the following sentence to the first paragraph of the Discussion:

“In future work it would also be informative to repeat the QTL mapping for variability in seed germination time under alternative conditions for maternal growth, seed storage and sowing, thus testing the extent to which common loci are involved under different environmental conditions.”

In the Results, it appears that the authors have conducted independent biological experiments of maternal growth and seed set. This provides the ability to directly quantify heritability of CV rather than the more visual analysis that has been done. The authors should utilize a linear model to directly estimate the heritability of CV in this trait to compare with other traits where CV heritability has been measured.

We thank the reviewer for this suggestion and have calculated the broad-sense heritability for CV. We have added:

“The broad-sense heritability (defined as the ratio of total genetic variance to total phenotypic variance) for CV was estimated to be ~40%, which is at the upper limit of heritabilities previously measured for variability in a number of post-germination plant traits (Hall et al., 2007)”

In Figure 1—figure supplement 1 part A/B, the authors should not include a line of 1:1 unless they have explicitly tested for a relationship. Given this analysis, the authors should conduct a mixed linear ANCOVA with genotype and seed age to directly quantify the influence of the two factors on the CV. Given this analysis, it is likely that they will find that the slope of CV is decreasing across storage time and as such, storage is having an influence on the CV. This will also allow them to show that the rank order of genotypes is not greatly changing across time.

The line showing x=y is for visualisation purposes and is not intended to imply a 1:1 relationship. We have added to the figure legend: “The black lines show x = y to aid data visualisation (they do not represent models fitted to the data)”.

We are not intending to claim that the CV is unchanged with days after ripening (DAR), but that the small deviation from x=y indicates that the CV obtained for a given line is not highly sensitive to this parameter.

We performed a mixed linear ANCOVA as the reviewer suggested, but it doesn’t reveal a specific pattern of change in CV with DAR. There is some variation in how each accession responds, but the uncertainty intervals are relatively large and mostly overlap with zero (no difference). This is plotted in Author response image 1, as the difference relative to 30 DAR (the data were standardised, so the units are standard deviations away from the overall mean):

**Author response image 1. sa2fig1:** 

In Figure 2B, it is not clear what these represent. Are these all the siliques on a single plant arranged from top to bottom or are they single siliques from the same plants shown in part A? Similarly, I'm not sure how the top and bottom half of a single silique provides sufficient information to claim that there is no variation along an inflorescence. And by bottom/top, is this distance from the maternal rosette or from the inflorescence stem? The gradients in a single silique would be largely driven by the distance from the inflorescence stem.

We have clarified these issues in the main text and in the legends of Figure 2 and figure 2—figure supplement 1. The figure shows different individual siliques from different plants, not all the siliques from one plant. The point is that variability can be found within one silique. We’ve updated the main text to make it clear that we’re not claiming that there is no difference between siliques depending on position, but that this wouldn’t be required to generate the variability in germination time we see for pooled seeds of the whole plant, as much of this variability is seen in individual siliques. The top/bottom comparison is presented to show that variation along the length of a silique (not along an inflorescence) is not what underlies the variability within a silique. Top/ bottom refers to the distance from the inflorescence stem, as the reviewer suggests, so we’ve clarified this. The main text states:

“To address whether the variability in germination time that we have observed can arise independently of between-silique differences, we collected seed from samples of individual siliques from 4 high or very high variability lines and characterised their germination time distributions. For these lines, the full range of germination times observed in whole-plant samples was also present in seed from individual siliques (Figure 2A, B; Figure 2—figure supplement 1). This suggests that variability in seed germination time can arise independently of position or age differences between siliques.

We next hypothesised that germination time might be related to the position of the seed along the longitudinal axis of the silique. To test this, we cut siliques into halves and sowed seeds from the top halves (i.e. distal halves, furthest from the mother plant’s inflorescence stem) and bottom halves separately. For the lines tested, late and early germinating seeds were produced by both halves of the siliques, with no consistent differences between the top and bottom halves of siliques in the fraction of seeds that germinated late (Figure 2C; Figure 2—figure supplement 1D). Thus, variability in germination time in the lines tested can arise independently of positional or maturation gradients within the whole plant or individual siliques. This suggests that a mechanism exists to generate differences in germination behaviour of equivalent seeds from the same silique, which is not dependent on gradients of regulatory molecules along the fruit.”

Figure 2 legend states:

B) As for A) but each row represents the distribution obtained using seeds from a single silique. Single siliques were randomly sampled from parent plants, and single siliques sampled from 7 parent plants are represented. C) Individual siliques were cut in half and seeds from the top and bottom halves (distal and proximal, furthest and closest to the mother plants’ inflorescence stems, respectively) were sown separately. Each row is the bottom and top half of a particular silique. Half siliques sampled from 2 parent plants are represented.

Figure 2—figure supplement 1 legend states:

“Germination time distributions for M182 (A) and M53 (B), for samples of seeds pooled from whole plants, or for single siliques separated into top and bottom halves. Seeds from whole plants and half siliques were obtained and sown in different experiments. The siliques were randomly sampled from parent plants, and halved siliques from multiple plants are represented (3 parent plants for M182 and 10 for M53). For whole plants, each row is the distribution obtained using a sample of pooled seeds from one plant. For half siliques, each row is the bottom and top half of a particular silique (top half is that furthest from the mother plants’ inflorescence stem). The size of the circles is proportional to the percentage of seeds that were sown that germinated on a given day. (C) As for (A) and (B) but for M4, showing whole siliques in the right-hand panel. Single siliques sampled from 8 parent plants are represented.”

As a general statistical guideline, whenever a regression line is shown in a plot, the figure legend should have the information surrounding the line, method, R2, N and significance. Additionally, most of these regressions are conducted using Pearson which is very sensitive to abnormal distributions which are present in most of these data. These should most likely be conducted using Spearman Rank or other similar non-parametric tests.

For Figure 3 and its supplements, we have removed the regression lines and calculated Spearman’s correlations, which we report in the figure panels in the following way: Spearman’s rho = , p = , n =.

I'm unsure why the authors correlated the parental effects to the haplotype effects for the two QTLs in a system where there is transgressive segregation and likely epistasis. The use of this correlation is only valid if the model is totally additive with no transgression. In this system, this data doesn't provide evidence one way or another. At best they could develop a two-locus model using the haplotypes and run them as a model together to look at additivity and epistasis to try and explain the parents. But running as single loci isn't informative.

The reviewer has a good point. If anything, the lack of correlation between QTL effect and parental phenotype does imply there is epistasis (the effect of the QTL depends on the background). We tried to fit the two-locus model as the reviewer suggests, but actually, this is not so straightforward. With 19 alleles in each locus, we have 19x19 possible genotype combinations, which is higher than our sample size of ~330 MAGICs. Basically, if we take combinations of Chr3+Chr5 haplotypes most of them occur only once. So, unfortunately, we can only really look at individual locus effects, not interaction (epistasis) effects, as the resulting uncertainty intervals are too wide.

We have instead replaced the panels in question with plots showing the ranked effects of the accessions’ haplotypes, not relating these to the accessions’ phenotypes:

The bet-hedging experiment seems unconnected to the natural variation. Essentially this treatment is a 100% selection event that is never going to occur in nature. The experiment is set up to test if non-germinated seedlings provide the ability to evade absolute death that occurs at a specific date prior to the latest germination time. As such, this is less an experiment and more a control that the phenotyping is reproducible. I'm not sure if this really adds to the manuscript as is. It would need to incorporate some aspect of the natural variation like are the high CV parents more likely to be in fluctuating environments, etc to help support the argument.

We have removed the bet-hedging experiment

Reviewer #2:If we consider the study in several consecutive parts such as A. The analysis on germination times of the diverse genetic backgrounds which led to B. a QTL analysis to find genes that were already known to influence germination, such as SOG1 and SOG6, that led to C. Modelling involving, very few by the way, players in ABAignalling pathways to explain D. that GA and ABA levels are key to understand phenotypic variation on germination time.1) My main concern is how can authors explain the need of the model in this story? since based on their data in B they could have gone directly to D.2) Moreover, the stochastic model suggested to test the levels of GA and ABA, but even if they sustain in their response the need of the model, how can they connect the data in B with the need to explore, involve, GA in the equation?3) Authors should elaborate how they justify the inclusion of GA signalling in the model, if QTL data did not suggest this. Of course, they can claim that GA and ABA are well known players in seed germination, but still the way it is written it feels that GA appears suddenly in the story.

We thank reviewer 2 for these comments and acknowledge that we did not previously do enough to justify the need for the model. As mentioned in the response to the editor’s point 1), we have now made a much closer link between the results of the QTL mapping and the modelling.

In response to reviewer 2’s comments, at the beginning of the manuscript we have now introduced the importance of the interactions between ABA and GA for the decision to germinate and emphasise that there is a lack of understanding about how the ABA-GA bistable switch could generate and influence variability in germination times. We now state:

“Pioneering modelling work has suggested that variable germination times can be generated by variation in sensitivities to germination regulators in a batch of seeds (Bradford, 1990; Bradford and Trewavas, 1994), or due to stochastic fluctuations in the regulators of ABA (Johnston and Bassel, 2018). Interestingly, a combination of experiment and modelling has revealed that the ABA-GA network can be described as a bistable switch due to the mutual inhibition between ABA and GA, leading to two possible states, a dormant high ABA low GA state, or a germinating low ABA high GA state (Topham et al., 2017). This regulatory motif can explain the observation that Arabidopsis seed germination is more effectively triggered by fluctuating temperatures than continuous cold (Topham et al., 2017). However, it is unclear whether different Arabidopsis genotypes generate different degrees of variability in germination times, or what the role of the ABA-GA bistable switch in generating these different germination time distributions might be.

Here we set out to investigate the causes of germination time variability between genetically identical Arabidopsis seeds and to explore how variability in germination time could be accounted for by the ABAGA bistable switch.”

We have also given more justification for the decision to model the genetic differences between lines with a difference in ABA sensitivity and state:

“There is evidence to suggest that the best candidate genes underlying our identified loci influence ABA sensitivity. The effect of the *DOG6* locus overlapping our chromosome 3 QTL is proposed to be caused by the *ANAC060* gene (Hanzi, 2014), which influences ABA sensitivity in seedlings, directly binds to the promoter of the ABA-responsive transcription factor, *ABA INSENSITIVE 5* (*ABI5*), and can down-regulate expression of both *ABA INSENSITIVE 4* (*ABI4*) and *ABI5* (Li et al., 2014; Yu et al., 2020). Coincidently, the two candidate genes for the chromosome 5 locus are closely related in function. The *SET1* locus in this region is hypothesised to be caused by the *ABA-HYPERSENSITIVE GERMINATION 1* (*AHG1*) gene (Footitt et al., 2019). *AHG1* is a type 2C protein phosphatase (PP2C) that inhibits ABA signalling via dephosphorylating class II SNF1-related protein kinase 2 (SnRK2), which promote seed dormancy by activating ABA-responsive transcriptional changes (Liu and Hou, 2018; Née et al., 2017a; Nishimura et al., 2007, 2018). DOG1 has been shown to directly bind to AHG1, independently from ABA, and inhibit its function, thus allowing DOG1 to inhibit germination via the ABA pathway (Carrillo-Barral et al., 2020; Née et al., 2017a; Nishimura et al., 2018). Mutants of all three candidate genes have altered ABA sensitivity: *anac060* mutant seedlings and *ahg1* mutant seeds have increased ABA sensitivity (Yu et al., 2020; Li et al., 2014; Nishimura et al., 2007), while *dog1* mutant seeds have decreased ABA sensitivity (Née et al., 2017a).” (This is followed by a characterisation of these mutants, see response to reviewer 2, comment 2).

And we give more justification for the requirement to include GA in the model:

“The hypothesis that natural variation in germination time distributions is caused by differences in ABA sensitivity raises the question of how differences in ABA sensitivity between lines could affect their levels of variability in germination time. To answer this, we built a simplified mathematical model of the core ABA-GA network that governs germination time (Liu and Hou, 2018) (Figure 5A). We reasoned that it was necessary to include both ABA and GA in the model, since the decision to germinate is governed by the relative levels of the two hormones (Née et al., 2017b; Shu et al., 2016b), and both converge to regulate expression of a common set of transcription factors that control seed dormancy and germination (Liu and Hou, 2018; Piskurewicz et al., 2008; Shu et al., 2016a). A previous modelling study that solely considered ABA regulation has proposed that stochastic fluctuations in the regulation of ABA can generate variability in germination times (Johnston and Bassel, 2018) and the ABA-GA network has been modelled previously to account for germination decisions (Topham et al., 2017). However, the ability of the ABA-GA network to generate variability in germination time has not been explored.”

Because of the cited previous work on the mechanistic link between ABA and GA, and the fact that the bistable switch that their interactions form is key to the decision to germinate, we believe that a model of ABA dynamics that doesn’t include GA would be significantly lacking. Additionally, the lack of strong GA-related candidate genes underlying the QTL we identified doesn’t mean that GA is not involved in the decision to germinate or in the mechanism underlying variability in germination time in the MAGIC lines, it only suggests that the main natural variation may be in ABA-related genes, but which are operating in the context of the ABA-GA network. A role for GA is supported by our results that modifying GA levels through exogenous addition (Figure 6—figure supplement 1) or genetically in GA biosynthesis mutants (Figure 7) affects the variability in germination time.

4) If the tracked phenotype is related to loci that involves SOG1, SOG6 and SET1genes, why authors did not generate, or use mutants in these genes to explore seed germination time? this is one of my major questions, and the only extra experiment work that this reviewer would like to see in a future version of the manuscript.

We thank the reviewer for this suggestion and in response we have characterised the germination time distributions and ABA sensitivities of the mutants. We found evidence that the 3 mutants tested (*dog1*, *ahg1* and *anac060*) influence ABA sensitivity and the distribution of germination times, with the strongest evidence for *dog1*. We have added the following to subsection “Effects of QTL candidate genes on seed germination time variability”:

“We tested these mutants for their effects on germination time distributions and found that the *dog1-3* mutant in the Col-0 background (Bentsink et al., 2006) consistently had reduced CV of germination time, and reduced mean and mode days to germination compared to the wild-type (Figure 4—figure supplement 5A). Since in Col-0 all seeds normally germinate within 3 days in seed batches stored for 30 days, we also did this experiment in seed batches stored for a shorter period of time (5 days after harvesting). In this case, in Col-0 there were later germinating seeds that were not present in the *dog1-3*mutant, making the effect of the *dog1-3*mutant allele more apparent. We also obtained a dog1 T-DNA insertion mutant in the No-0 accession background (Kuromori et al., 2004), since the No-0 haplotype of the Chr5 QTL locus is predicted to be associated with high variability in the MAGIC lines (Figure 4—figure supplement 2A). The dog1 mutant in the No-0 background showed a similar phenotype to dog1-3 in Col-0, with reduced CV, mean and mode (Figure 4—figure supplement 5C). Consistent with previously published work (Née et al., 2017a), we observed reduced ABA sensitivity of the dog1 mutants in the Col-0 and No-0 background in our germination assays (Figure 4—figure supplement 5B, D). On the other hand, the ahg1-5 mutant in the Col0 background (Née et al., 2017a) showed a slight increase in CV of germination time, which was associated with an increase in ABA sensitivity (Figure 4—figure supplement 5E, F). Thus, both of these candidate genes for the Chr5 locus have an effect on the CV of germination time which is consistent with their altered ABA sensitivities.

For the anac060 mutant in the Col-0 background (Li et al., 2014), we observed a slight increase in CV, with an increase in mode and decrease in percentage germination compared to Col-0 in seed sown 3 days after harvest (Figure 4—figure supplement 5G, 3 DAH), but did not see a convincing phenotype in seeds that were stored for 30 days prior to sowing (i.e. in the same conditions as the MAGIC lines used for QTL mapping) (Figure 4—figure supplement 5G, 30 DAH). We observed a weak tendency towards an increase in ABA sensitivity in the anac060 mutant, but this was not as striking as that reported in seedlings (Li et al., 2014). Thus, it is possible that a gene other than ANAC060 underlies the Chr3 QTL, or if it is responsible, then its effect on germination may depend on the genetic background or on the specific alleles present in the populations we studied.

Overall our results support the hypothesis that the candidate genes underlying the Chr 5 QTL peak could influence variability through an effect on ABA sensitivity and suggest that this could also be the case for the ANAC060 candidate gene for the Chr 3 QTL peak.”

In these cases, each shade of grey is the germination time distribution from a single experiment, with the behaviours of its 3 separate seed batches pooled together for plotting, treated as one sample.

We discuss these results in the Discussion, by stating:

“Further work will be needed to identify the causal variants underlying both of the QTL identified here and to determine the extent to which they have correlated or uncoupled effects on different germination traits. It will be important to test directly candidate genes through transforming candidate gene null mutants with corresponding alleles cloned from high and low variability lines (Weigel, 2012). These transgenic lines could then be used to understand how natural variation at the two QTL influences germination time distributions, by using them to quantify the effects of the different accession alleles on CV, mode and percentage germination as well as on traits related to the ABA-GA network such as ABA and GA sensitivity.”

Reviewer #3:In this manuscript, Abley and colleagues map differences in germination phenotypes, including the coefficient of variation (CV), the mode, and the percent germinated, in the *A. thaliana* MAGIC line population, finding two QTL. They then develop a computational model for germination incorporating the integration of ABA and GA pathways to generate a bistable switch for the germination decision and demonstrate that the model can output a variety of CV, mode, and germination percentage phenotypes. Using the model, they then predict and test the response of MAGIC lines with high and low CV to exogenous ABA and GA applications.I agree with the general premise that bet-hedging and CV variation are potentially ecologically and evolutionarily important and I appreciate the development of a model that serves as a basis for understanding both the molecular and natural genetic mechanisms that generate germination phenotypes. My major concern with this manuscript, however, is that it doesn't quite meet its promise in showing that the bistable switch proposed by the model underlies natural variation in the phenotypes measured here. What is demonstrated is that the model can (possibly) account for the effect of endogenous ABA or GA on MAGIC lines with different CVs. Strictly speaking this isn't the same as demonstrating that the model explains how this CV variation in the MAGIC (or parental lines) is generated to begin with. To do this, one would have to have specific information/predictions about how underlying genetic variation affected model components and then demonstrate that THESE modulations could generate the CV variation observed in parental lines (or MAGIC lines). For example, let's say that a particular allele at the QTL on chromosome three caused reduced expression of DELLAs, subsequently changing the integrator parameter of the model. Would incorporating this modified parameter explain the CV variation between plants with the different variants of this QTL? In some cases, the manuscript makes the distinction between the response and origin questions clear, but in key areas (like the title and abstract), this difference gets blurred.

We thank the reviewer for their suggestion to make a more explicit connection between the genetic variation and the model, which we have taken on board. See response to Editor’s comment 1.

1) The experiment meant to demonstrate that there is no pattern of germination variability along the plant is not very conclusive. Showing single silique distribution patterns with siliques ordered basipetally would more convincingly demonstrate that there is no relationship between silique age and CV.

Please see response to reviewer 1 comment 9. We have changed the main text to make it clear that we’re not claiming there is no relationship between silique age/ position and CV, but that since individual siliques show a large degree of variability in germination times, germination distributions that we see in seeds pooled from whole plants can be accounted for independently from differences in germination time across the plant. Our results from single and half siliques show that a mechanism exists to generate differences in germination time between seeds in the same developmental context.

2) I am not convinced that mode, CV, and percent germination can be genetically uncoupled. For the most part, there are fairly strong relationships between these phenotypes in the MAGIC lines, with a few (maybe 10 or so) outlying lines where the CV is (presumably) much higher than would be predicted from the mode or percentage germination (Figure 4A – by the way, what is the overlap between these outliers and with the lines with bimodal CV distributions?). The data demonstrating uncoupling consists of several cherry-picked examples in Figure 3 (and its associated supplements). One would really need to look at this more systematically to demonstrate the extent to which this decoupling is actually significant (i.e. considering the variability in mode and CV estimates between replicates, are there lines in which the relationship between CV and mode is different from the predicted relationship?). And in fact, the QTL results also suggest that these phenotypes can't be uncoupled – any QTL for mode or percent is also a QTL for CV, and the QTL on chromosome 5 that the authors claim is CV-specific both explains little variation and its marginal significance depends on whether or not 8 particular lines with bimodal distributions (and therefore perhaps cases in which mode doesn't describe centrality well) are included in the analysis. Even in the bulk segregant analysis, there is a clear relationship between germination time of F2s parents and CV of subsequent F3s. I therefore think that the strength of this claim needs to be reconsidered.

We thank the reviewer for pointing this out, please see response to Editor’s comment 2. We have changed the text throughout so that now we described the traits are being weakly correlated rather than decoupled. In response to reviewers’ specific questions, there is still some degree of uncoupling between CV and mode when the bimodal lines are excluded from the data, for example, from Figure 3A ii, where only lines with CV < 0.6 are shown (and therefore bimodal lines are excluded), lines with a mode of 3 show a range of CVs between ~0.1 and ~0.6. Therefore, we do not agree that it is only in cherry picked examples where variation in CV independent of variation in mode is seen. We do agree however that the traits are most likely genetically and mechanistically linked and we have changed how we present the QTL mapping in accordance with the reviewer’s suggestion, deemphasising a specific effect of the Chr5 QTL on CV. As mentioned in the response to editor’s comment 2, we now state in the Discussion:

“In the QTL mapping, the chromosome 5 locus was only found to be significantly associated with the CV of germination time. However, besides their described effect on CV, the mutant alleles of the two candidate genes we tested, DOG1 and AHG1, also affected mode and percentage germination (Née et al. 2017; Bentsink et al. 2006). The fact that we did not detect an association between this region of chromosome 5 and those two latter traits could be explained by differences between the naturally occurring alleles within the MAGIC population (which may predominantly affect CV) and the mutant alleles of these genes (which affect multiple traits). It could also be the case that the MAGIC lines’ alleles affect multiple germination traits but that we didn’t have enough statistical power to detect this.”

In relation to the question about Figure 4A “what is the overlap between these outliers and with the lines with bimodal CV distributions?”

This is explained in the main text, where we state: “However, there are 8 lines that tend to have bimodal distributions when sown on agar (e.g. Figure 1A, M182 and M178). As such, these lines lie at the extreme tail of the distribution of CVs, with much higher values than the other lines (Figure 1B). Therefore, we ran our QTL scans both with and without the bimodal lines, as their extreme values may affect the QTL results disproportionately (Figure 4A, B).”

3) Comparing Figure 6 and Figure 7 (the predicted and observed responses of GA and ABA in high and low CV lines), there seem to be quite a number of cases in which the predicted response is different from that observed (the mean and mode in response to GA for example) or in which there is extremely high variability among the lines tested, with only a subset of lines matching the predicted responses (the CV and mean response of high CV lines to ABA, for example). So at what point can one say that the predicted response is consistent with the observed one? And why would one expect such high variability between lines? Do they harbor QTL alleles that would explain these differences? This might be one way to tie the mapping and modeling parts of the manuscript together better (they currently kind of feel like two disconnected stories – one doesn't need to have mapped QTL to generate the model or confirm the response of high/low CV lines to GA and ABA).

We agree that there is variation between the different high/ low variability lines, and this cannot be explained by their haplotypes at the Chr3 and Chr 5 QTL (We have now addressed this in the text, please see response to reviewer 1, comment 1 for details). However, we believe that this situation is to be expected given the genetic differences between the lines throughout the genome. It is likely that other loci which we were not able to detect in the QTL mapping underlie their different responses to the hormones. We have now raised this possibility in the Discussion:

“We found that other parameters caused a variety of other behaviours in terms of the relationship between the CV and mode of the germination time distributions. This is consistent with the observation of a complex relationship between CV and mode in the MAGIC lines and accessions, whereby these traits are weakly positively correlated between lines, but can be partially decoupled. Since it is likely that we have not detected all genetic loci that influence germination time distributions in our QTL mapping (the Chr 3 and Chr 5 QTL together only account for 23% of the variance in CV in the MAGIC lines), it is possible that undetected genetic variation in multiple components of the ABA-GA network contributes to the variation in germination time distributions that we observe and the complex relationship between CV, mode and percentage germination in the data.”

In response to the specific points:

- “there seem to be quite a number of cases in which the predicted response is different from that observed (the mean and mode in response to GA for example)”,

In both model and experiment the mean and mode tend to decrease in response to GA, but the magnitude of this response is greater in the model. We have now plotted both simulation results and experiments on a log scale and the results look more similar than previously. The model has not been parameterized to fit the data in any way, therefore we believe that a qualitative match in behaviour is the best that can be expected. We conclude: “Thus, for both ABA and GA, the overall effects of exogenous addition were qualitatively similar between the model and experiments.” We are only claiming that a qualitative comparison can be made.

- “or in which there is extremely high variability among the lines tested, with only a subset of lines matching the predicted responses (the CV and mean response of high CV lines to ABA, for example)”.

We have changed the text to mention the variation amongst the lines tested:

“For high variability lines, increasing concentrations of ABA increased mean and mode germination times, but had relatively little effect on CV (Figure 6B i, ii and iii, Figure 6—figure supplement 4A, 182). Some high variability lines show an increase in CV followed by a decrease, which is consistent with the model, and some lines show a slight decrease (Figure 6B i). It is possible that these lines occupy different positions in parameter space due to variation at loci (and therefore components of the ABA-GA network) that are not accounted for in our simulations of these lines. Consistent with a less striking effect of exogenous ABA on the CV of high variability lines compared to low variability lines, in the model, the fold-changes in CV upon ABA addition for high variability lines were smaller than those predicted for low variability lines (Figure 6A i).”

4) The experiment meant to demonstrate the adaptive value of bet hedging is quite trivial – one doesn't really need an experiment to show that killing more plants in one genotype will result in fewer plants of that genotype – and therefore I think the importance of result is overstated in the manuscript. Demonstrating that differences in CV have adaptive value really would be a difficult experiment to do and would need to incorporate some sort of natural (or very closely simulated natural) conditions. If such experiments are attempted, I would also strongly recommend NOT using Col as one of the accessions. Col is presumably adapted to lab conditions, not natural ones, and it is easy to think of how early and low CV germination would be selected for in a laboratory setting (especially if CV can be impacted by a 1-2 major loci).

We have removed the bet-hedging experiment.